# Halide-assisted differential growth of chiral nanoparticles with threefold rotational symmetry

Jiapeng Zheng[1,2,8], Christina Boukouvala[3,4,8], George R. Lewis[3,4,8], Yicong Ma[5], Yang Chen[1], Emilie Ringe [3,4] ✉, Lei Shao [6] ✉, Zhifeng Huang[7] & Jianfang Wang [1,2] ✉

Enriching the library of chiral plasmonic nanoparticles that can be chemically mass-produced will greatly facilitate the applications of chiral plasmonics in areas ranging from constructing optical metamaterials to sensing chiral molecules and activating immune cells. Here we report on a halide-assisted differential growth strategy that can direct the anisotropic growth of chiral Au nanoparticles with tunable sizes and diverse morphologies. Anisotropic Au nanodisks are employed as seeds to yield triskelion-shaped chiral nanoparticles with threefold rotational symmetry and high dissymmetry factors. The averaged scattering $g$-factors of the L- and D-nanotriskelions are as large as 0.57 and −0.49 at 650 nm, respectively. The Au nanotriskelions have been applied in chiral optical switching devices and chiral nanoemitters. We also demonstrate that the manipulation of the directional growth rate enables the generation of a variety of chiral morphologies in the presence of homochiral ligands.

The exploration of chirality has been extended from molecules and organisms to nanophotonics and other interdisciplinary studies, with the development of chiral plasmonics that enable strongly enhanced asymmetric light–matter interaction at the nanoscale by making use of surface plasmon resonance[1,2]. Emerging colloidal chiral plasmonic nanoparticles that can be chemically synthesized have therefore been attracting increasing attention since the recent development of their synthetic methods[3–5]. The seed-mediated chiral-ligand-directed growth method, which is based on the reduction of noble metal precursors on achiral seeds, has emerged as a promising strategy to produce nanoparticle enantiomers that support strong chiroptical properties[6]. A set of chiral ligands and specific seeds eventually yield chiral structures, such as 432 helicoid nanoparticles and helicoid

nanorods[3,4,7,8]. Such chiral plasmonic nanoparticles have been explored in enhancing the contrast of valley-dependent photoluminescence of transition metal dichalcogenide monolayers[9], triggering asymmetric photocatalysis[10], and modulating immunological reactions[5]. Despite the recent encouraging progress in the synthesis of chiral plasmonic metal nanoparticles, fine control of their morphology, size, and chirality has remained a challenge.

A universal method for synthesizing chiral plasmonic nanoparticles would greatly expand their applications in biology, chemistry, and physics. Traditional seeded growth methods generally adopt anisotropic seeds or exploit anisotropic growth schemes to obtain high flexibility in morphology regulation[11,12]. For example, anisotropic seeds such as Au nanorods and nanoplates can experience a variety of

[1]Department of Physics, The Chinese University of Hong Kong, Shatin, Hong Kong SAR, China. [2]Shenzhen Research Institute, The Chinese University of Hong Kong, Shenzhen 518057, China. [3]Department of Materials Science and Metallurgy, University of Cambridge, Cambridge CB3 0FS, United Kingdom. [4]Department of Earth Sciences, University of Cambridge, Cambridge CB2 3EQ, United Kingdom. [5]Department of Physics, Hong Kong Baptist University, Kowloon Tong, Hong Kong SAR, China. [6]State Key Laboratory of Optoelectronic Materials and Technologies, Guangdong Province Key Laboratory of Display Material and Technology, School of Electronics and Information Technology, Sun Yat-sen University, Guangzhou 510275, China. [7]Department of Chemistry, The Chinese University of Hong Kong, Shatin, Hong Kong SAR, China. [8]These authors contributed equally: Jiapeng Zheng, Christina Boukouvala, George R. Lewis. ✉e-mail: er407@cam.ac.uk; shaolei5@mail.sysu.edu.cn; jfwang@phy.cuhk.edu.hk

evolutions into nanoparticles with unimaginable morphologies[13–15]. Halides such as iodide and bromide ions exhibiting preferential adsorption on specific facets help to trigger the formation of noble metal nanoparticles with a variety of shapes[16]. Such strategies can also be employed to control the growth of chiral nanoparticles and may contribute to the production of chiral noble metal nanoparticles with rich morphologies and tunable optical activities but have not been explored.

Considering the role of anisotropic seeds and the effect of halides on anisotropic growth, we propose a growth strategy of halide-assisted differential growth (HADG) on anisotropic metal nanoparticle seeds for the generation of plasmonic metal nanocrystals with shape chirality (Fig. 1). HADG enables high-degree control over the morphology, size, and chiroptical properties of the generated chiral plasmonic metal nanocrystals. Au nanodisks, composed of the {111} crystal facets on the top and bottom and mixed crystal facets on the side, were selected as the anisotropic seeds (Supplementary Fig. 1 and Supplementary Table 1)[17]. The evolution of the Au nanodisks into three-dimensional (3D) chiral nanocrystals involves the reduction of the Au precursor in the presence of glutathione (GSH), ascorbic acid (AA), cetyl-trimethylammonium bromide (CTAB), and potassium iodide (KI) (Fig. 1a). The introduction of KI and CTAB in the growth solution causes different growth rates along different directions and changes the growth pathway of the chiral nanocrystals (Fig. 1b and Supplementary Figs. 2–4). The geometric models constructed from scanning electron microscopy (SEM) characterization show that each type of the obtained chiral nanocrystals has distinct chiral shapes. They, therefore, exhibit different chiroptical properties. Based on the differential growth of the anisotropic nanodisks, we established nanostructure libraries consisting of a series of chiral metal nanocrystals and achiral nanogears (Fig. 1c–f). These nanocrystals were obtained by adjusting the amounts of CTAB and KI that are frequently employed during the seed-mediated growth of noble metal nanocrystals in the presence of GSH, without the need of adding other complex chiral molecular ligands or introducing additional external stimuli such as stirring vortexes and circularly polarized laser pulses[4,5].

## Results

### Morphology and optical properties of Au nanotriskelions

We first prepared chiral Au nanocrystals consisting of a triple spiral with rotational symmetry which we refer to as nanotriskelions (Fig. 2). SEM imaging and 3D electron tomography analysis were performed using high-angle annular dark-field scanning transmission electron microscopy (HAADF-STEM) to characterize the chiral morphology of the Au nanotriskelions with different handedness. Electron tomography shows that there are three twisted arms extending from the center of the {111} facets, forming the triskelion-shaped wrinkles with threefold rotational (TFR) symmetry (Fig. 2a–f and Supplementary Movies 1, 2). Hausdorff chirality measurements reveal that the Au nanocrystals prepared in the presence of L- and D-GSH have triskelion-shaped wrinkles with counterclockwise and clockwise rotational directions, respectively (Supplementary Fig. 5)[18].

The Au nanotriskelions with excellent helicoidal morphology were synthesized in the presence of Au nanodisks, GSH, AA, CTAB, and KI. We then studied the role of the reagents on the growth of the nanotriskelions. The helicoidal morphology of the Au nanotriskelions comes from the generation of high-Miller-index facets on the Au nanodisks. Previous works have demonstrated that the formation of chiral nanorods with a 422 symmetry arises from the stabilization of the {521} facets[19]. The reducing agent AA is the key to the generation of high-Miller-index facets on the Au nanodisks such as the {541} facets (Supplementary Fig. 6)[20]. CTAB and KI are employed as growth regulators. Iodide ions favorably bind to the {111} facets and reduce the Au deposition rate on the {111} facets[13,14]. Bromide ions, on the other hand, are preferentially adsorbed on the {100} facets[21]. The directional growth along the <100> and <111> directions can be blocked with increased CTAB and KI concentrations, respectively. We note that chloride ions cannot be used to control the chiral growth of the Au

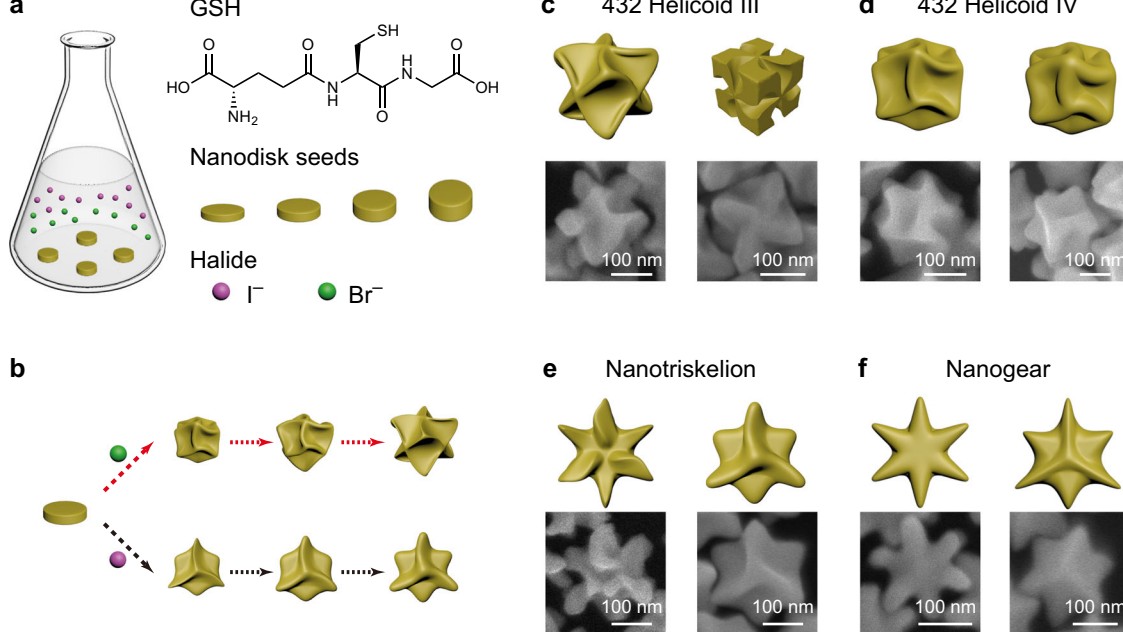

**Fig. 1 | Schematic illustrating the synthesis process of the chiral nanocrystals.** **a** The as-prepared Au nanodisks were added as the seeds into the growth solution made by mixing GSH, CTAB, KI, HAuCl₄, and AA. **b** Chiral nanocrystals in a variety of shapes prepared by HADG on the anisotropic nanodisks. The Au nanodisks can evolve into chiral nanocrystals with different dominant rotation symmetries by using different CTAB/KI concentrations. **c**–**f** SEM images and schematics of the typical chiral and achiral nanocrystals, including 432 helicoid nanocrystals with dominant fourfold rotational (FFR) symmetry (**c**, **d**), Au nanotriskelions with dominant threefold rotational (TFR) symmetry (**e**), and nanogears (**f**). These nanocrystals were synthesized using the Au nanodisks as the seeds and different CTAB/KI concentrations.

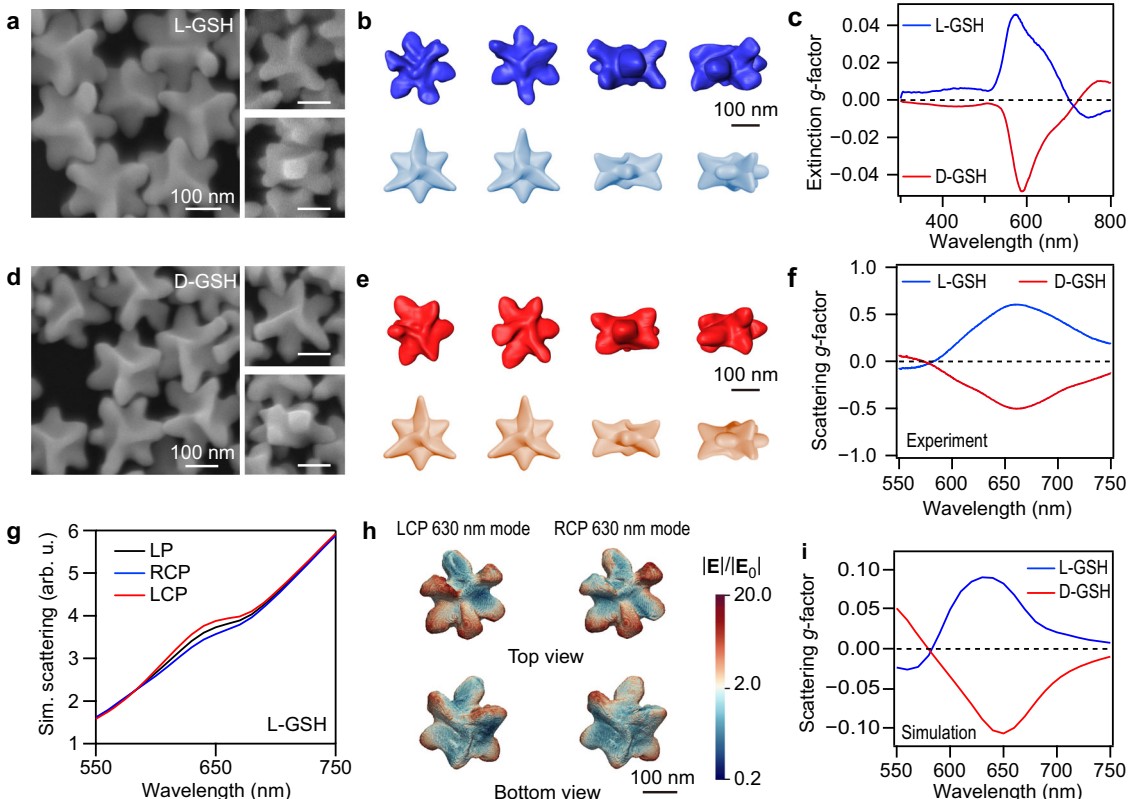

**Fig. 2 | Morphology and chiroptical properties of the Au nanotriskelions. a** SEM images of the Au L-nanotriskelions. **b** Tomography reconstruction (top) and constructed model (bottom) of the L-nanotriskelions. **c** Dissymmetry factor spectra measured from CD spectrometry for the L- and D-nanotriskelions in solution. **d** SEM images of the Au D-nanotriskelions. **e** Tomography reconstruction (top) and constructed model (bottom) of the D-nanotriskelions. The Au nanotriskelions in (**a**, **d**) were grown from the 90/30 nm (diameter/thickness) nanodisks in the presence of L- and D-GSH. All the scale bars in (**a**, **d**) are the same. **f** Average scattering g-factor

spectra obtained from the CDS measurements on the L- and D-nanotriskelions. **g** Discrete dipole scattering simulation of the CDS of the L-nanotriskelions. The scattering intensities excited by the linearly polarized (LP), LCP, and RCP light were calculated. **h** Electric fields of the L-nanotriskelion under the excitation of LCP and RCP light at 630 nm. The distributions of the electric field enhancement are shown in the color maps at the logarithmic scale. **i** Simulated scattering g-factor spectra of the typical L- and D-nanotriskelions. Source data are provided as a Source Data file.

nanodisks because chloride ions show lower binding affinities on the Au surface than the bromide and iodide ions (Supplementary Fig. 7)[22]. GSH enantiomers are the chiral directors for the growth of the chiral nanocrystals with opposite handedness (Supplementary Figs. 6, 7). Similar chiral structures can also be produced when GSH is replaced by cysteine (Supplementary Fig. 8). Compared with cysteine, GSH with a larger molecular weight can interact with more kink atoms[3]. When the growth of the nanotriskelions was guided by GSH, larger protrusions can be produced from the nanodisks, leading to the nanotriskelions with an improved optical chirality.

The helicoidal morphology of the nanotrikelions results in excellent chiroptical properties. The extinction circular dichroism (CD) spectra of the Au nanotriskelions were measured on a CD spectrophotometer. The extinction dissymmetry factor (g-factor) was obtained to quantify the chiroptical behavior of the nanocrystals in solution, as described by ref. 3

$$g = 2 \times \frac{A_{LCP} - A_{RCP}}{A_{LCP} + A_{RCP}} \quad (1)$$

where $A_{LCP}$ and $A_{RCP}$ are the extinction spectra for the left-handed circularly polarized (LCP) and right-handed circularly polarized (RCP) incident light, respectively. The opposite rotation directions of the arms on the L- and D-nanotriskelions lead to their opposite asymmetric interaction with circularly polarized light, as proved by the inverted CD activities at ≈590 nm (Fig. 2c). However, the Au nanotriskelions with strong anisotropic shape in solution are randomly orientated, giving

rise to a low absolute value of the g-factor. Circular differential scattering (CDS) measurements were therefore performed on the individual Au nanotriskelions lying on substrates (Supplementary Fig. 9)[23,24]. The scattering spectra of the individual nanoparticles were measured under the excitation of LCP and RCP light ($S_{LCP}$ and $S_{RCP}$). The scattering dissymmetry factor ($g_S$) was calculated by

$$g_S = 2 \times \frac{S_{LCP} - S_{RCP}}{S_{LCP} + S_{RCP}} \quad (2)$$

We have measured the CDS spectra for a number of nanotriskelions, including 41 L-type and 32 D-type ones. The averaged scattering g-factors of the L- and D-nanotriskelions are as large as 0.57 and −0.49 at 650 nm, respectively (Fig. 2f). The geometry obtained from the HAADF-STEM tomography was used for the discrete dipole scattering (DDSCAT) simulation to investigate the relationship between the chiral structure and the chiroptical response measured from CDS spectroscopy (Fig. 2g–i). The plasmonic mode of the L-nanotriskelion at 630 nm was easier to be excited by LCP light than by RCP light (Supplementary Figs. 10, 11).

A combination of CD and CDS measurements is helpful in comprehensively revealing the relation between the helicoidal structure and the chiroptical response of chiral plasmonic nanoparticles. We next studied the differences in structural and chiroptical responses between the 432 helicoid III and Au nanotriskelions (Supplementary Figs. 12, 13). The 432 helicoid III nanocrystals with strong chiroptical properties were synthesized from Au octahedrons and are of a cubic

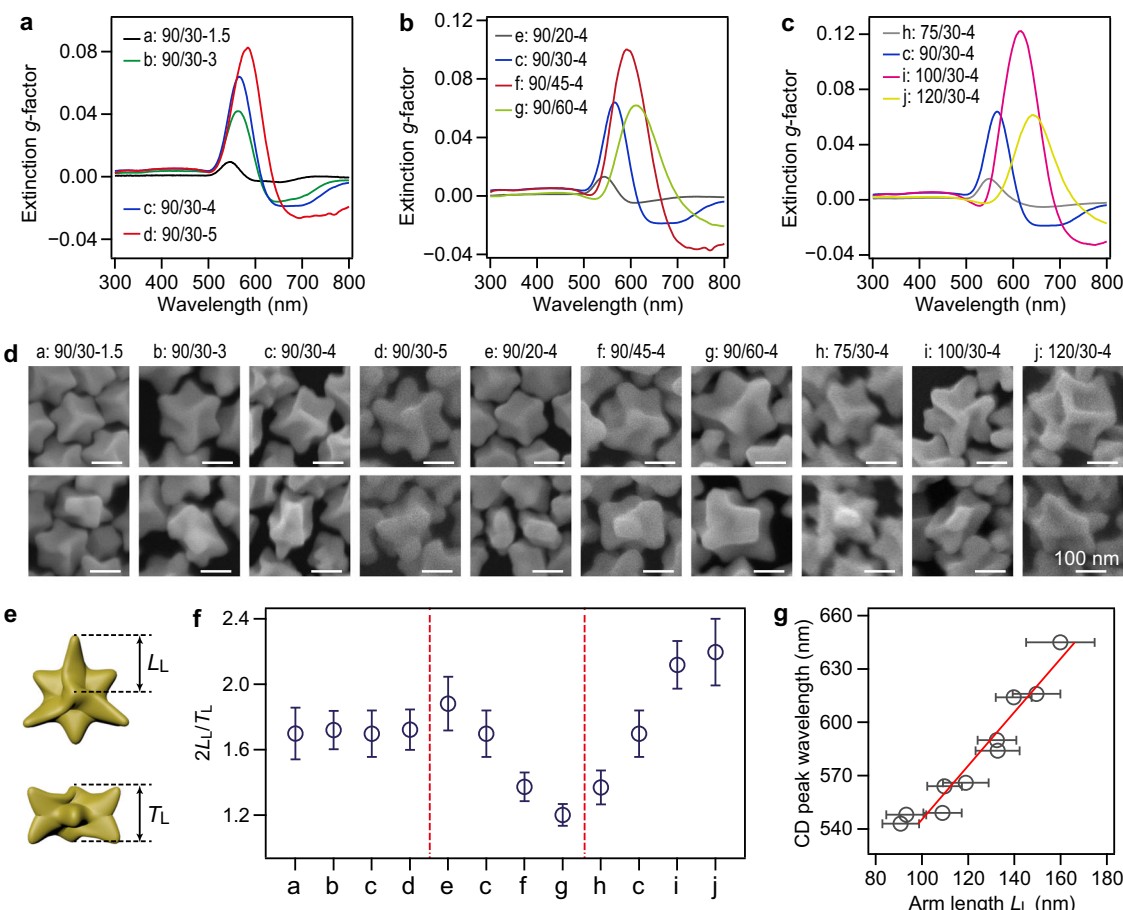

**Fig. 3 | Controllable synthesis of the Au L-nanotriskelions. a–c** Extinction dissymmetry factor spectra for the L-nanotriskelions with various dimensions, which were synthesized by use of the Au precursor with different amounts (**a**) and the Au nanodisk seeds with different thicknesses (**b**) and diameters (**c**). The nanotriskelions marked by, e.g., 90/20-4, means that the seeds are the Au nanodisks with an average diameter of 90 nm and an average thickness of 20 nm and the added amount of the Au precursor is 4 μmol for the Au nanodisk seeds with an optical density of 0.8. **d** SEM images for the L-nanotriskelions with varying dimensions, including the observation from the top (top) and side (bottom). All the scale bars are the same. **e** Schematics of the Au nanotriskelions with the arm length $L_L$ and the thickness $T_L$ indicated. **f** Aspect ratios of the Au nanotriskelions, defined as $2L_L/T_L$ in (**d**). All error bars show mean ± standard deviation. $n = 100$ independent experiments. **g** Dependence of the peak wavelength in the CD spectra on the nanotriskelion arm length. The trend is represented as the red solid lines. All error bars show mean ± standard deviation. $n = 100$ independent experiments. Source data are provided as a Source Data file.

geometry[3,25]. Their six faces exhibit pinwheel-like patterns that consist of four highly curved arms of increasing widths. The strong chiroptical properties of the 432 helicoid III and Au nanotriskelions stem from their chiral geometries, which dominantly exhibit fourfold rotational (FFR) symmetry along the <100> directions and TFR symmetry along the <111> directions, respectively. Although the 432 helicoid III nanocrystals (grown from octahedrons) and the Au nanotriskelions (grown from nanodisks) are prepared in the presence of GSH with the same handedness, the nanotriskelions viewed from the <111> directions and the cubic helicoid nanocrystals viewed from the <100> directions exhibit opposite geometrical chirality (Supplementary Fig. 12). As a result, they exhibit opposite chiroptical responses in the wavelength range of 500−700 nm. For example, Supplementary Fig. 13 shows that the L-432 helicoid III exhibits an extinction $g$-factor of −0.088 at 590 nm and the average scattering $g$-factor is −0.45 at 640 nm. On the other hand, the L-nanotriskelions (Fig. 2c, f) show an extinction $g$-factor of 0.05 at 595 nm and the average scattering $g$-factor is 0.57 at 650 nm.

## The controllable synthesis of Au nanotriskelions

Gold nanotriskelions with increased dimensions and enlarged chiral surfaces were synthesized by increasing the amount of the Au precursor in the growth solution or using Au nanodisks of increased thicknesses and diameters as the seeds (Fig. 3, Supplementary Figs. 14,

15, and Supplementary Table 2). The spectral evolution of the extinction $g$-factor displays the redshifted CD band for the Au nanotriskelions with increased dimensions (Fig. 3a–d and Supplementary Figs. 14a–d, 15). The Au nanotriskelions synthesized from the nanodisks naturally show anisotropic structures (Fig. 3e and Supplementary Fig. 14e). The aspect ratio of the nanotriskelions was defined as $2L_L/T_L$ or $2L_D/T_D$, with $L$ and $T$ representing the arm length and thickness, respectively (Fig. 3f and Supplementary Fig. 14f). The nanodisks with higher diameter-to-thickness aspect ratios yield the nanotriskelions with higher aspect ratios. The plasmonic CD bands of the obtained nanotriskelions shift toward longer wavelengths with increasing arm lengths (Fig. 3g and Supplementary Fig. 14g). We can therefore obtain the Au nanotriskelions with large aspect ratios and strong chiroptical response. For example, the L-Au nanotriskelions grown from the 100/30 nm (diameter/thickness) nanodisks show a high aspect ratio at ≈2.1 and display enhanced chiroptical response with an extinction $g$-factor of 0.12 at 616 nm (Supplementary Table 2). We note that the Au nanodisks with diameter-to-thickness aspect ratios ranging from 1.5 to 4.5 are the key to triggering the anisotropic structure of the nanotriskelions. Other seeds with larger aspect ratios, such as Au triangular and hexagonal nanoplates, can evolve into nanoparticles with dendritic morphologies (Supplementary Fig. 16). Additionally, the HADG strategy guarantees the mass-production synthesis of the Au

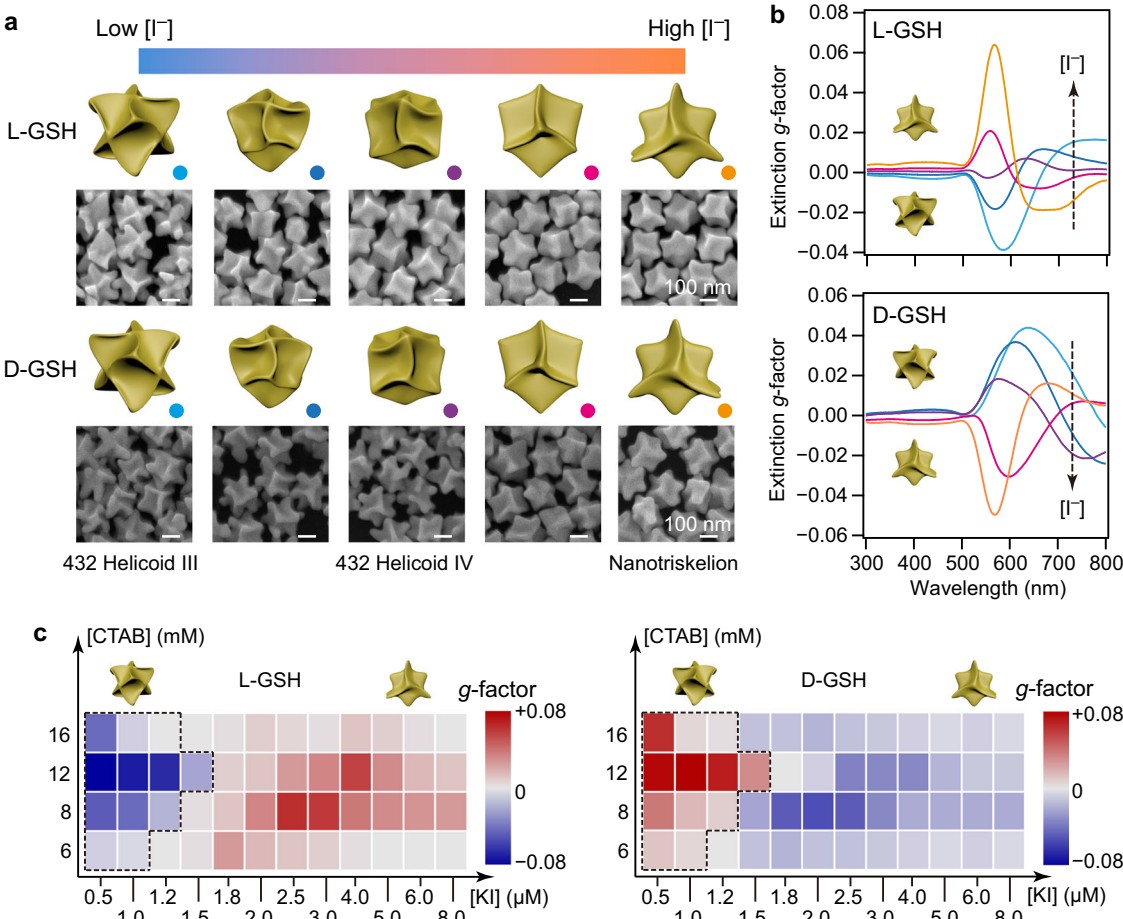

**Fig. 4 | Morphology and chirality evolution of the chiral nanocrystals grown from the Au nanodisks by HADG.** The Au nanodisks with a size of 90/30 nm and the diameter-to-thickness ratio of 3.0 were employed as the seeds. **a** Effect of the KI concentration on the morphology of the obtained chiral nanocrystals. The 432 helicoid III nanocrystals are formed at low I⁻ concentrations (0.5–1.0 μM) and the 432 helicoid IV nanocrystals start to appear as the I⁻ concentration is increased. Further addition of KI (1.8–8.0 μM) finally results in the generation of the Au nanotriskelions. The morphology evolution was confirmed by SEM imaging. The

scale bars in all the SEM images are the same. **b** Spectral evolution of the extinction g-factors for the chiral nanocrystals in (**a**). The chirality inversion of the optical response demonstrates that the grown chiral nanocrystals change from the 432 helicoid III nanocrystals to the Au nanotriskelions. The dashed lines show the spectral evolution with increasing KI concentrations. **c** Dependence of the extinction g-factors of the grown chiral nanocrystals on the concentrations of CTAB and KI. The dashed frames show the grown chiral nanocrystals with FFR symmetry. Source data are provided as a Source Data file.

nanotriskelions with a yield larger than 80% (Supplementary Fig. 17a–f). The volume of the growth solution for the Au nanotriskelions can be enlarged to 120 mL, much larger than that for previously reported chiral nanoparticles, which are usually limited to a few milliliters because of the involved complex chiral ligands or additional external stimuli[4,5]. The Au nanotriskelions also show good stability. The morphology of the nanotriskelions can be preserved well for at least 6 months when they are stored in an environment of 0–5 °C (Supplementary Fig. 17g–i).

**HADG strategy for the synthesis of various chiral nanoparticles**

In addition to the Au nanotriskelions, we also demonstrated that the Au nanodisks can also evolve into the 432 helicoid III and IV nanoparticles (Fig. 4 and Supplementary Figs. 2–4) that were reported to grow from highly isotropic nanoparticle seeds in previous studies[3,7]. The concentration of I⁻ was adjusted during the HADG process. We believe that the opposite chiroptical responses of the 432 helicoid III nanocrystals and the nanotriskelions should result from their different geometries, which are controlled by the introduction of halides during the seeded growth. The halides modulate the growth rates along the <100> and <111> directions of the Au nanocrystals differently.

The overgrowth through HADG to produce different chiral Au nanocrystals is largely affected by the halide ion concentration as well as the size and shape of the starting seeds. We first varied the concentrations of KI and CTAB, and thus the concentrations of I⁻ and Br⁻, in the growth solution during the HADG process to prepare chiral nanocrystals with different rotational symmetries (Fig. 4). When the growth was guided by L-GSH, the 432 helicoid III nanocrystals and the Au nanotriskelions show negative and positive chiroptical response at 500–700 nm, respectively. The 90/30 nm Au nanodisks were found to successively evolve into 432 helicoid III, 432 helicoid IV, and nanotriskelions with increasing KI concentrations (Fig. 4a, also see Supplementary Figs. 2–4). The extinction CD activity is transformed from negative to positive. When the growth was guided by D-GSH, we observed the opposite trend of the chirality evolution with increasing KI concentrations (Fig. 4b). Considering the twinned structures of the Au nanodisks, we conjecture that the 432 helicoid III and IV nanoparticles grown from the nanodisks have twinned structures[26], compared with the single-crystalline 432 helicoid nanoparticles grown from Au octahedrons[3]. The chiral nanocrystals oriented along the <100> and <111> directions are summarized in Supplementary Fig. S18. We can see that the dominant symmetry changes from FFR to TFR with increasing KI concentrations. The correlograms on the chirality

evolution show that the evolution maintains within the CTAB concentration range of 6–16 mM (Fig. 4c and Supplementary Fig. 19). The cubic 432 helicoid III nanocrystals always exhibit opposite chiroptical response in the extinction CD spectra to that of the nanotriskelions. Optimized CTAB/KI concentrations can thus be determined from experiments to ensure the largest dissymmetry in the optical response for both types of chiral nanocrystals.

We then constructed a library of the 432 helicoid III and Au nanotriskelion enantiomers grown from the Au nanodisks with varying sizes (Fig. 4 and Supplementary Figs. 20–25) (90/30 nm, 90/45 nm, 90/60 nm, and 100/30 nm). A variety of the CTAB/KI concentrations were employed during the growth directed by both types of GSH enantiomers. Large-scale SEM imaging and artificial identification were used to pick up the dominant morphology of the grown nanocrystals. The 3D frameworks of the obtained nanocrystals were thereafter reconstructed from the SEM images taken along different crystallographic orientations.

The growth from thicker nanodisks (90/45 nm and 90/60 nm) showed a similar evolution tendency with that when the 90/30 nm nanodisks were employed as the seeds. The nanodisk seeds preferentially grow into the chiral 432 helicoid nanocrystals over a large CTAB/KI concentration range (CTAB: 8–20 mM, KI: 0.5–2.0 µM) (Supplementary Figs. 20–23), while the synthesis of the Au nanotriskelions requires higher concentrations of CTAB and KI. The 100/30 nm nanodisks, on the other hand, are difficult to evolve into the 432 helicoid nanocrystals. Instead, they prefer to grow into various nanotriskelions at a wide CTAB/KI concentration range (CTAB: 8–20 mM, KI: 1.2–8.0 µM) (Supplementary Figs. 24, 25). The morphology and chiroptical properties of the chiral nanocrystals obtained from the optimized CTAB/KI concentrations when the Au nanodisks of different sizes (90/30 nm, 90/45 nm, 90/60 nm, 100/30 nm) were employed as the seeds are summarized in Supplementary Fig. 26 and Supplementary Table 3. The 90/45 nm nanodisks can evolve into nanotriskelions and 432 helicoid III nanocrystals of the strongest chiroptical properties with absolute values of the extinction g-factors larger than 0.1.

Most Au nanodisks with different sizes (90/30 nm, 90/45 nm, and 90/60 nm) can grow into the 432 helicoid III nanocrystals and nanotriskelions with an opposite chiroptical response by adjusting the concentrations of CTAB and KI. The result is against the perception that nanoparticles with opposite chiroptical response must be grown by use of ligand enantiomers with opposite handedness[3,7]. Our HADG process may therefore have a different chiral nanocrystal growth mechanism from previous reports that the achiral seeds first grow into highly faceted structures as an intermediate state, followed by the generation of twisted geometries according to the enantioselective interaction between the chiral ligands and the chiral facets[3].

As shown above, when the Au nanodisks are used as the seeds, the evolution of the grown chiral nanocrystals exhibits a complex dependence on the concentrations of CTAB and KI. Such dependence is highly determined by the aspect ratio of the Au nanodisk seeds. The Au nanodisks with the diameter-to-thickness ratio at 3.3 (more anisotropic) prefer to grow into Au nanotriskelions while the Au nanodisks with the diameter-to-thickness ratios at 1.5 (more isotropic) tend to give rise to cubic helicoid nanocrystals. To reveal the growth mechanism, we investigated the chiral growth on Au octahedron seeds that were encapsulated with the {111} facets. The octahedrons with edge lengths of 55 nm were found to steer the generation of the 432 helicoid nanocrystals. As the KI concentration is increased in the growth solution, the resultant chiral nanocrystals evolve from 432 helicoid III to IV, exhibiting blue-shifted CD bands and attenuated CD response (Supplementary Fig. 27). The 432 helicoid nanocrystals become more and more cubic with increasing CTAB concentrations (Supplementary Fig. 28). Au nanotriskelions, in contrast, failed to be synthesized from these isotropic seeds.

A HADG mechanism of the chiral Au nanocrystals is proposed according to the above experimental observations. The crystallographic structure of the seeds determines the preferential growth position, and the added halide selectively controls the growth rates along specific crystalline directions. The growth direction can therefore be adjusted isotropically for the octahedral seeds enclosed by the eight {111} facets and regulated anisotropically for the Au nanodisks whose thickness and diameter determine the area ratio of the {111} facets to the other facets. The chiral molecular ligands further bias the growth into one mirror image of the chiral nanocrystal. As a result, the nanodisks with large aspect ratios at 3.3 preferentially develop into Au nanotriskelions while those with small aspect ratios at 1.5 tend to evolve into 432 helicoid nanocrystals. The anisotropic growth rate control brought by halide ions is critical in generating the chiral nanocrystals with different shapes, which was confirmed by imaging the morphologies of the chiral nanocrystals at different growth stages (Supplementary Figs. 29–31). The time-dependent study on the growth dynamics proved that KI and CTAB at the increased concentrations can reduce the growth rates along the <111> and <100> directions, respectively.

HADG on large aspect-ratio Au nanodisks resulted in an exceptional morphological diversity of the obtained Au nanotriskelions. A library of Au nanotriskelions with various surface profiles were therefore constructed (Supplementary Figs. 25, 32 and Supplementary Table 4). When the 100/30 nm nanodisks were employed as the seeds, iodide ions at the increased concentrations blocked the growth along the <111> directions of the Au nanocrystals, leading to the lateral shrinking of the twisted arms on the nanodisks. On the other hand, CTAB at the increased concentrations slowed the growth rate along the <100> directions and changed the lateral contour of the nanotriskelions from hexagram to triangle. Further increase of the KI concentration led to the synthesis of achiral gear-shaped nanocrystals (Supplementary Figs. 32, 33). We believe that the formation of such nanogears is due to the fact that KI at high concentrations ([KI] >10 µM) can inhibit the evolution of chiral surfaces and simultaneously maintain the structure contour. Straight arms are preferentially generated from the {111} facets of the Au nanodisks. The nanogear morphology can also be tuned by increasing the CTAB concentration to slow down the growth of the in-plane arms.

## Au nanotriskelions for chiroptical switching and emissions

We further demonstrated the use of the Au nanotriskelions as a platform to realize different nanophotonic functionalities. The assembly of the chiral nanoparticles onto substrates can exhibit strong optical anisotropy and produce a special response to incident light. These features enable chiral plasmonic substrates potentially as spatial light modulators[25], enantioselective sensors[27], and other chiral nanophotonic devices. When the Au nanotriskelions are randomly and densely coated on a substrate, we found that over 85% of the Au nanotriskelions can lie on the substrate with their chiral surface facing the incident light. The anisotropic geometric morphology and chirality enable the Au nanotriskelions deposited on substrates to display a stronger CD response than the nanocrystals in solution (Supplementary Fig. 34). When the chiral nanoparticle density is increased, more chiral structures are involved in the absorption and scattering of the incident light, resulting in the improved chiral response (Supplementary Fig. 35). A chiral optical switching device was thus fabricated by depositing the nanotriskelions randomly and densely on a substrate and coating the obtained nanocrystal array with an active electrochromic polymer layer that can be chemically and electrochemically switched between different states (Fig. 5a–c and Supplementary Fig. 36)[28,29]. The obtained active chiral optical substrate exhibited a large and reconfigurable extinction CD peak shift over 100 nm when the coated polyaniline (PANI) was switched

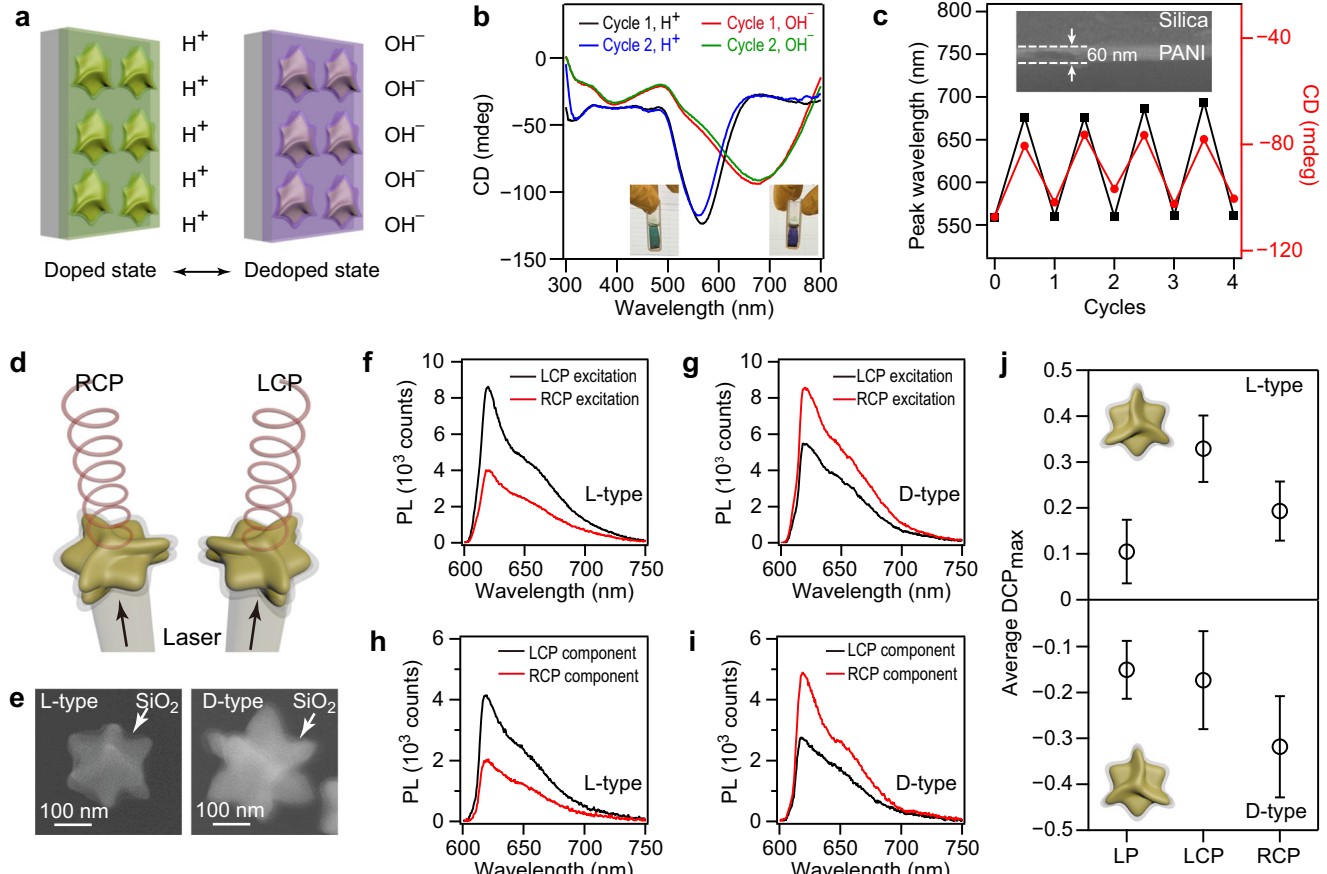

**Fig. 5 | Au nanotriskelions as a platform to realize chiroptical switching and emissions. a** Chiroptical switching device. The Au nanotriskelion-deposited substrate is covered with a PANI layer. Switching can be realized by proton-doping and dedoping when the substrate is immersed in aqueous HCl (pH = 2) and NaOH (pH = 11) solutions, respectively. **b** CD spectra of a D-nanotriskelion-deposited substrate coated with a 60 nm PANI layer during the proton-doping and dedoping processes in two cycles. Insets: photographs of the substrate immersed in HCl (left) and NaOH (right). **c** Reversible modulation of the peak wavelength and peak CD value in the CD spectra over five cycles of the proton-doping and dedoping processes. Inset: cross-sectional SEM image of the substrate showing the thickness of the PANI layer. **d** Chiral PL enhanced by the Au nanotriskelions. The fluorophore molecules are embedded in a silica shell surrounding the chiral Au nanocrystal. **e** SEM images of the L- and D-nanotriskelions coated with silica shell that is embedded with R640 molecules. **f, g** PL spectra of the hybrid nanostructures under LCP and RCP excitation. The L-type nanostructures show higher PL intensities under LCP excitation (**f**) while the D-type nanostructures display stronger PL when excited by the RCP laser light (**g**). **h, i** Polarization-resolved PL spectra of the chiral nanostructures under the differently polarized excitation. The L-type nanostructures show higher PL intensities in the LCP component when excited by the LCP laser (**h**) while the D-type nanostructures exhibit stronger PL intensities in the RCP component when excited by the RCP laser (**i**). **j** Maximum degree of circular polarization (DCP$_{max}$) values from the L-type and D-type nanostructures. All error bars show mean ± standard deviation. $n \geq 10$ independent experiments. The handedness of PL polarization shows a close relationship with the chiroptical response of the nanotriskelions. Source data are provided as a Source Data file.

during its proton-doped and dedoped states. Peak shifts as large as 200 nm were also achieved in the active chiral substrate made by the L-nanotriskelion-deposited substrate coated with a 130 nm PANI layer (Supplementary Fig. 36). The modulation of the chiroptical response with excellent stability and reversibility can be potentially applied in switchable polarizers.

In addition to the chiroptical switching device, we also demonstrated the feasibility of constructing nanoemitters that preferentially emit LCP or RCP photons from the individual Au nanotriskelions. Rhodamine 640 (R640), showing strong absorption at 575 nm and emission at 620–650 nm, was selected as the organic fluorophore for chiral emission[30]. (Gold nanotriskelion)–fluorophore hybrid nanostructures were prepared by embedding R640 into the shell of the silica-coated nanotriskelions (Fig. 5d–j and Supplementary Fig. 37). The thickness of the silica shell was 10–15 nm. The Au nanotriskelions can largely scatter and absorb CP light at 590–650 nm. Under the excitation at 594 nm, the interaction between the excitons in R640 and the plasmonic chiral near-field triggers plasmon–exciton coupling that is sensitive to the polarization of the excitation laser light (Fig. 5f, g).

The emitted photons can be scattered by the Au nanotriskelions, resulting in enhanced chiral emissions. Under the excitation of LCP and RCP laser light, the core@shell nanostructures exhibit largely different photoluminescence (PL) intensities, suggesting that the chiral nanocrystals modify the excitation efficiency (up to ≈40% in the emission anisotropy factor) of photons with different circular polarization states (Fig. 5h, i). The chiral nanotriskelions were also found to enable the emissions of partially circularly polarized photons when the fluorophores were excited by linearly polarized light, with the emission anisotropy factor reaching 20% (Fig. 5j). The emission polarization modulation can be further optimized by rationally designing nanotriskelion-based plasmonic structures to introduce stronger near-field optical chirality. Such emission polarization modulation of nanoemitters will be highly useful for the construction of nanophotonic circuit elements for future on-chip communication applications.

## Discussion
Using homochiral molecules to guide the growth of inorganic crystals and realize the diversity of chiral structures is of great significance.

A universal method is employed to prepare a library of plasmonic chiral nanocrystals without the requirement of external stimuli or specific complex reagents during the synthetic process. We believe that the developed HADG method can be used to guide the chiral growth on other Au nanoseeds. In nature, homochiral L-amino acids are used to transfer molecular chirality into biogenic crystals[31], but the underneath mechanism is still under debate. Recent discoveries suggest that the complex morphologies of biogenic crystals come from the anisotropic growth on symmetry-related facets[32]. The HADG method follows a similar strategy and can even result in the production of chiral metal nanocrystals of high purity and uniformity with nearly continuously varying geometries, implying that we can mimic nature by employing the differential growth as a general means for the fine and precise control of chiral growth.

## Methods

### Materials
Gold chloride trihydrate (HAuCl₄·3H₂O, 99.9%), cetyltrimethylammonium chloride (CTAC, 98%), trisodium citrate (TSC, 99%), sodium dodecyl sulfate (SDS, 99%), sodium borohydride (NaBH₄, 98%), L-ascorbic acid (AA, 99%), hydrochloric acid (HCl, 5 M), aniline (99%), potassium persulfate (KPS, 99%), hydrogen peroxide (H₂O₂, in water, 30 wt%), sodium hydroxide (NaOH, 99%), L-cysteine (L-Cys, 99%), D-cysteine (D-Cys, 99%), and L-glutathione (L-GSH, 98%,) were purchased from Sigma-Aldrich. Cetyltrimethylammonium bromide (CTAB, 98%) was obtained from Alfa Aesar. Tetraethyl orthosilicate (TEOS, 98%) was purchased from Acros Organics. Rhodamine 640 perchlorate (R640) was purchased from Exciton·Luxottica. D-glutathione (D-GSH, 99%) was purchased from Apeptide. Potassium iodide (KI, 99%) was purchased from Aladdin Reagent. Deionized water with a resistivity of 18.2 MΩ cm and produced by a Direct-Q 5 ultraviolet water purification system was used throughout all experiments.

### Synthesis and purification of the triangular Au nanoplates
To prepare the nanodisk seeds, triangular Au nanoplates were first prepared and then grown into hexagonal nanoplates[33]. The circular Au nanodisks were obtained through anisotropic oxidation on the hexagonal Au nanoplates (Supplementary Fig. 1). The triangular Au nanoplates were synthesized by a three-step seed-mediated growth method. The seed solution was prepared by adding TSC (0.01 M, 1 mL) and HAuCl₄ (0.01 M, 1 mL) solutions into water (36 mL), followed by the injection of a freshly prepared, ice-cold NaBH₄ solution (0.1 M, 1 mL) under vigorous stirring. The resultant solution was then aged in an isothermal oven at 35 °C for 2–6 h. Three solutions were prepared in three vials labeled A, B, and C. The solutions in vials A and B were prepared by adding CTAB (0.1 M, 4.5 mL), NaOH (0.1 M, 0.05 mL), and KI (0.01 M, 0.05 mL) into water (4.5 mL). The solution in vial C was made by mixing CTAB (0.1 M, 45 mL), NaOH (0.1 M, 0.5 mL), and KI (0.01 M, 0.5 mL) with water (45 mL). 0.05, 0.05, and 0.5 mL of AA (0.1 M) were introduced into vials A, B, and C, respectively. The solution color in the three vials then turned colorless, resulting from the reduction of the Au(III) species into Au(I) species. Next, the seed solution (1 mL) was added to the solution in vial A under gentle shaking. The resultant solution in vial A (1 mL) was then added into vial B under gentle shaking for ≈3 s. All of the solutions in vial B was then transferred to vial C under gentle shaking. The obtained mixture in vial C, with a volume of ≈100 mL, was left undisturbed in an isothermal oven at 35 °C for 24 h. Triangular Au nanoplates were thus produced and precipitated into the bottom of vial C. Spherical Au nanoparticles were also generated as impurities in the supernatant of vial C. To obtain high-purity triangular Au nanoplates, the supernatant was gently poured out after storage for 24 h, and water (20 mL) was then added to vial C. The resultant solution in vial C showed a green color, containing purified triangular Au nanoplates with sharp corners. The triangular Au nanoplates were

stored at room temperature and the sharp corners gradually became rounded.

### Synthesis of the hexagonal Au nanoplates
The triangular Au nanoplates with the major localized surface plasmon resonance (LSPR) peak at ≈800 nm were employed as seeds for the preparation of the hexagonal Au nanoplates. The optical density of the triangular Au nanoplate solution at the major plasmon peak was adjusted to be 3.0 (optical path length of 1.0 cm) by adding water into the as-grown sample. The growth solution was prepared by adding HAuCl₄ (0.01 M, volume $v_1$), AA (0.1 M, volume 0.5$v_1$), and water sequentially into a CTAB solution (0.1 M, 25 mL). The volume of the added HAuCl₄ solution $v_1$ was set at 3.0, 4.5, 7.5, and 10.0 mL to obtain hexagonal Au nanoplates with thicknesses of 20, 30, 45, and 60 nm, respectively. Water was added to adjust the total volume of the growth solution to 200 mL. The triangular Au nanoplate seed solution (50 mL) was then added to the growth solution under stirring. The resultant mixture was kept in an isothermal oven at 35 °C for 10 h.

### Synthesis of the circular Au nanodisks
The hexagonal Au nanoplates were washed by centrifugation, redispersed in CTAB solution (0.01 M, 250 mL), and then mixed with HCl (1 M, 2 mL) and H₂O₂ (35 wt%, 200 μL) solutions. The resultant mixture was gently stirred and treated with a water bath heating at 60 °C. The Au atoms at the corners and edges of the hexagonal nanoplates were etched by gentle oxidation because the CTAB molecules adsorbed at such positions are loosely packed. This oxidation process led to the formation of circular Au nanodisks. During the oxidation process, the diameter gradually decreased while the thickness remained unchanged. The extinction spectra of the mixture were monitored every 0.5 h and the blueshift of the major LSPR peak was observed. The oxidation could be terminated through high-speed centrifugation for the removal of the oxidant, resulting in Au nanodisks with desired sizes. The resultant Au nanodisks were then redispersed in CTAB solution (1 mM). Through this process, we produced circular Au nanodisks with diameters/thicknesses of about 75/30, 90/30, 100/30, 120/30, 90/20, 90/45, and 90/60 nm, and their LSPR peaks at 590, 600, 615, 628, 640, 580, and 560 nm, respectively. The sizes and LSPR peak wavelengths of the nanodisk samples were summarized in Supplementary Table 1. The optical densities of the Au nanodisk solutions at the major plasmon peak were then adjusted to be 0.8 (optical path length of 1.0 cm) by adding water.

### Synthesis of the Au octahedrons
The Au octahedrons were synthesized by a seed-mediated growth method[34]. The seed solution was prepared by mixing CTAC (0.2 M and 5 mL) and HAuCl₄ (0.5 mM and 5 mL) solutions, followed by injection of a freshly prepared, ice-cold NaBH₄ solution (0.02 M and 0.45 mL) under vigorous stirring. The resultant solution turned brown immediately and was aged in an isothermal oven at 35 °C for 2 h. Two growth solutions were prepared in two vials labeled D and E. The solution in vial D was prepared by adding CTAC (0.2 M and 5 mL), HAuCl₄ (0.01 M and 250 μL), and KI (0.01 M and 5 μL) into water (4.465 mL). The solution in vial E was made by mixing CTAC (0.2 M and 250 mL), HAuCl₄ (0.01 M and 12.5 mL), and KI (0.01 M and 250 μL) solutions with water (223.25 mL). 0.22 and 11 mL of AA solution (0.1 M) were introduced into vials D and E, respectively. The solution color in the two vials then turned colorless. The seed solution (60 μL) was next added into the solution in vial D under gentle shaking until the solution color turned light pink. The resultant mixture (3 mL) in vial D was then added into vial E under gentle shaking for ≈10 s. The obtained mixture was left undisturbed at room temperature for 24 h. The resultant solution in vial E showed a cherry-red color, containing high-purity Au octahedrons. The obtained Au octahedrons were then redispersed in CTAB solution (1 mM) after centrifugation. We adjusted the optical density of

the Au octahedron solution at the plasmon peak to be 0.8 (optical path length of 1.0 cm) by adding water.

## Synthesis of the chiral nanocrystals

Most of the chiral nanocrystals described in this work were synthesized through HADG on the Au nanodisks. The growth solution was prepared by first adding CTAB, KI, L-GSH (2.75 mM, 120 μL) or D-GSH (2.75 mM, 160 μL), and HAuCl₄ solution into water sequentially. After the addition of AA (0.1 M and 1 mL) into the growth solution, the mixture solution turned colorless. The Au nanodisk seed solution (2 mL) was then rapidly injected into the mixture solution under stirring. The resultant mixture, with a total volume of 10 mL, was kept in an isothermal oven at 35 °C for 2 h. The CTAB/KI concentrations in the growth solution were varied to obtain different chiral and achiral nanocrystals. In most cases, 0.4 mL HAuCl₄ (0.01 M) solution was used in the growth solution. HAuCl₄ solutions at different amounts were used in the synthesis of the Au nanotriskelions with different dimensions (Fig. 3 and Supplementary Figs. 14, 15). For the cysteine-directed synthesis of the Au nanotriskelions (Supplementary Fig. 8), 600 μL of L-Cys (2 mM) and D-Cys (2 mM) solutions were used to replace the L- and D-GSH solutions. For the synthesis of the 432 helicoid nanocrystals from the Au octahedron seeds, a similar procedure was employed and L-GSH (2.75 mM and 150 μL) and D-GSH (2.75 mM and 180 μL) solutions were used. All the other growth conditions of the chiral nanocrystals were summarized in Supplementary Tables 2–4.

## Preparation of the (chiral nanocrystal)–fluorophore hybrid nanostructures

The (chiral nanocrystal)–fluorophore hybrid nanostructures were prepared through the generation of a fluorophore-embedded silica shell coated on the Au nanotriskelions. The fluorophore solution was first prepared by dissolving R640 (50 mg) in ethanol (5 mL) and then mixed with the precursor for silica coating. The silica precursor solution was made by mixing TEOS (1 mL) with ethanol (4 mL). The prepared Au nanotriskelions, which were washed for the removal of the residual CTAB molecules and then redispersed in water (5 mL), were subsequently mixed with NaOH (0.1 M and 50 μL) and CTAB (0.1 M and 80 μL) solutions. The fluorophore-containing silica precursor (50 μL) was then dropped into the nanotriskelion-containing solution every 2 h under stirring. After stirring for 10 h, a total of 250 μL of the silica precursor solution was added. A silica shell was thus generated on the nanotriskelions through the hydrolysis of the silica precursor. R640 molecules were simultaneously embedded in the mesostructured silica shell. The hybrid nanostructures were finally washed twice and redispersed in water.

## Deposition of the Au nanotriskelions densely on substrates

Glass slides or indium tin oxide (ITO)-coated glass substrates were first cleaned by ultrasonication in ethanol for 30 min, and then treated with oxygen plasma for 3 min. The as-prepared Au nanotriskelion solution (10 mL) was first centrifuged, and the precipitate was redispersed in CTAB solution (0.01 mM, 1 mL). The concentrated solution was kept at room temperature for 24 h. Most Au nanotriskelions were precipitated into the bottom of the centrifuge tube. The supernatant (950 μL) was then gently taken out and thrown away, with 50 μL of the residual solution remaining. Water (950 μL) was then added to the residual solution, with the CTAB concentration becoming ≈5 μM. The cleaned glass slide or ITO substrate was then immersed into the resultant Au nanotriskelion solution for at least 12 h. The Au nanotriskelions were as a result deposited densely onto the substrate through electrostatic interaction between the CTAB-stabilized positively charged nanocrystals and the negatively charged substrate. The Au nanotriskelions were randomly positioned, forming large-area and high-density monolayers on the substrate. The deposition density can be controlled by the deposition time. The Au nanotriskelion-deposited

substrate was taken out from the nanocrystal solution, rinsed with water, and finally blown dry with nitrogen for further use.

## Fabrication of the chiral optical switching devices

The chiral optical switching device was achieved through the construction of active chiral optical substrates by coating PANI that can be switched between different states on the Au nanotriskelions deposited on substrates[28]. The substrate deposited with the high-density nanotriskelion monolayer was first immersed in CTAB solution (1 mM and 20 mL) for 10 min and then rinsed with water. The substrate was next immersed in SDS solution (4 mM and 20 mL) for 4 h. The deposited Au nanotriskelions were therefore covered with SDS molecules through electrostatic interaction between SDS and CTAB molecules. The precursor solution for PANI deposition was prepared by mixing aniline solution (0.1 M and 2 mL) with HCl (pH = 3.0, 30 mL). KPS solution (70 mM and 1.6 mL) was then added into the precursor solution under stirring for 5 min. The Au nanocrystal-deposited substrate was then immersed into the mixture and kept for 1 h. The mixture solution turned light green, indicating the formation of PANI. The sample was thereafter rinsed with water and blown dry with nitrogen. The above process can be repeated for different cycles to control the thickness of the formed PANI layer. One, two, and four circles of PANI deposition resulted in PANI layer thicknesses of 20, 60, and 130 nm, respectively. We also achieved the coating of PANI through an electrochemical means by employing a typical three-electrode system in an electrochemical workstation (CHI 760E)[35]. A standard Ag/AgCl reference, a Pt wire, and the Au nanotriskelion-deposited ITO substrate functioned as the reference, counter, and working electrodes, respectively. All the electrodes were immersed in the electrolyte solution containing HNO₃ (2.0 M) and aniline (0.1 M). After a constant potential at +0.80 V was applied for different periods of time, PANI was produced on the ITO substrate. To measure the chiroptical switching performance, the substrate deposited with the PANI-covered nanocrystals was immersed into HCl (pH = 2) and then NaOH (pH = 11) solutions for different circles.

## Electron microscopy

SEM images were taken on a JEOL JSM 7800 F microscope at an operation voltage of 10 kV. TEM imaging and selected-area electron diffraction analysis were performed on an FEI Tecnai Spirit 12 microscope at an operation voltage of 120 kV. HRTEM and HAADF-STEM imaging were carried out on an FEI Tecnai F20 microscope equipped with an Oxford energy-dispersive X-ray analysis system.

## TEM tomography

The TEM samples were prepared by drop-casting the aqueous nanocrystal suspension (10 μL) onto a holey carbon grid. The data were acquired on an FEI Krios operated at 300 keV. For each sample, 100 HAADF-STEM images were acquired at steps of 1.5° between ±60° and steps of 1° from ±60° to ±70°. The frames obscured at high angles were removed manually. Image-shift alignment was then performed on Sobel-filtered images using a phase-correlation algorithm and tilt-axis alignment using a manual procedure to minimize arcing in the reconstructed orthoslices. Tomographic reconstruction was performed using a compressed sensing algorithm formulated with a weighting of 0.05 for the total variation regularization term[36], implemented with 1000 iterations of a Chambolle-Pock algorithm[37]. Intensity thresholding and segmentation were then carried out in Avizo to produce the final isosurfaces.

## Extinction and CD spectroscopy

The extinction spectra of the plasmonic nanocrystals were measured on an ultraviolet/visible/near-infrared spectrophotometer (Perkin-nElmer Lambda 950). The extinction CD spectra of the plasmonic

nanocrystals were measured on a CD spectrophotometer (JASCO J-1500).

## SEM-correlated single-particle CDS spectroscopy

The Au nanotriskelions were deposited on cleaned Si substrates with a 300-nm-thick $SiO_2$ layer. An optical system for single-particle CDS measurements was established based on an Olympus microscope (BX53). The light generated from a quartz-tungsten-halogen lamp (100 W) was adjusted to pass through a linear polarizer (Union Optic, 550–900 nm) and a quarter-waveplate (Union Optic, 550–750 nm). The circularly polarized light then passed through a 100× dark-field objective (Olympus, NA 0.9) to create annular excitation with an incidence angle of 64°. The scattered photons were collected through the same objective and directed to a spectrometer (Acton SpectraPro 2360i) connected to a liquid-nitrogen-cooled charge-coupled-device camera (Princeton Instruments, Pixis 400, cooled to −70 °C). The measured scattering spectrum of a nanoparticle was corrected by first subtracting the background spectrum taken from the adjacent region without any nanoparticle and then dividing the obtained spectrum with the pre-calibrated response curve of the entire optical system. A pattern-matching method was employed to capture the same nanoparticle on the SEM and scattering images. The scattering response of an individual nanoparticle can therefore be correlated with its morphology.

## PL spectroscopy

The Au nanotriskelions coated with the fluorophore molecules were prepared to study the coupling between the chiral plasmons and the excitons. An experimental setup was built for circular polarization-resolved PL spectroscopy. A 594 nm laser was employed for excitation. The laser was linearly polarized or circularly polarized before reaching the nanoparticle. A set of linear polarizer and a quarter-waveplate were placed at the spectrometer entrance for the measurement of the LCP and RCP components of the PL signal. A 600 nm long-pass filter was employed to block the excitation laser light. To quantitatively evaluate the chiral emissions from the nanoparticle, the degree of circular polarization (DCP) was calculated according to

$$DCP = \frac{PL_{LCP} - PL_{RCP}}{PL_{LCP} + PL_{RCP}} \tag{3}$$

where $PL_{LCP}$ and $PL_{RCP}$ are the intensities of the LCP and RCP PL components, respectively. We also measured the maximal DCP values ($DCP_{max}$) and peak wavelengths for more than ten nanostructures (Supplementary Fig. 37) and calculated the average $DCP_{max}$ values under the excitation of LP, LCP, and RCP laser light.

## Hausdorff chirality measurement

In order to quantify the handedness and extent of geometrical chirality, we calculated a Hausdorff chirality metric[38]. This was achieved by first generating the idealized 3D reference models of the left- and right-handed nanotriskelions based on their appearance in SEM (Supplementary Fig. 5a). Each tomography reconstruction (Supplementary Fig. 5b) was then aligned as closely as possible to the left- and right-handed reference models. The mean Hausdorff distance was calculated for both (mean over all points on the reconstructed shape of the distance from each to its closest neighboring point in the reference shape) using MeshLab[39]. The Hausdorff distance is a measure of similarity between two shapes, with a lower distance implying a closer match. We obtained the L-to-D Hausdorff distance ratios of 0.72 and 1.15 for the nanotriskelions synthesized from L- and D-GSH respectively, confirming that the synthesis gives rise to the intended chirality.

## Electromagnetic simulation

The optical scattering spectra and electric field distributions were obtained numerically using DDSCAT[40,41], an open-source code that solves Maxwell's equations in the discrete dipole approximation approach. The tomography-reconstructed shapes were exported in .stl files, which were subsequently converted into the required dipole array (Supplementary Fig. 10a) for DDSCAT using a freely available MATLAB function[42]. The electric field distributions were plotted using the ParaView application. The frequency-dependent dielectric constant of metallic Au was taken from the data of Johnson and Christy and the ambient refractive index was set to 1.0[43]. For linearly polarized light, the average of two orthogonal polarizations was taken. Calculations were carried out with ≈400,000 dipoles, which gave an interdipole distance of 0.1 nm. This is significantly small under the consideration of the complexity of the nanotriskelion geometry and on the basis of convergence tests (Supplementary Fig. 10b). The scattering g-factors were further calculated, as described by Eq. (2).

## Reporting summary

Further information on research design is available in the Nature Portfolio Reporting Summary linked to this article.

## Data availability

The data that support the findings of this paper are available from the corresponding authors upon request. Source data are provided with this paper.

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

## Acknowledgements

C.B. is thankful for funding from the Engineering and Physical Sciences Research Council (EPSRC, Standard Research Studentship (DTP) EP/R513180/1). G.R.L. is thankful for funding from the EPSRC NanoDTC Cambridge (EP/L015978/1). L.S. acknowledges support from the Pearl River Talent Recruitment Program (2019QN01C216). J.W. acknowledges support from the Croucher Foundation (Croucher Senior Research Fellowship 2020-2021), the Hong Kong Government (Research Matching Grant Scheme, Project Code 8601434), and the Research Grants Council of Hong Kong (ANR/RGC, A-CUHK404/21). This project received support from Shenzhen Science and Technology Program (JCYJ20210324140805014) and the European Research Council (ERC) under the European Union's Horizon 2020 research and innovation program (grant agreement No. 804523).

## Author contributions

J.Z., L.S., E.R., and J.W. conceived the project. J.Z. performed the nanocrystal syntheses, electron microscopy characterization, CDS measurements, model construction, chiral substrate fabrication, and photoluminescence measurements. C.B. and G.R.L. performed the data analyses for STEM tomography, Hausdorff chirality measurements, and electromagnetic modeling. J.Z., Y.M., and Z.H. performed CD measurements. J.Z. and Y.C. fabricated the active chiral substrates. J.Z., C.B., G.R.L., L.S., E.R., and J.W. wrote the manuscript with comments from all authors. J.W., E.R., and L.S. supervised the project.

## Competing interests

The authors declare no competing interests.
