## [Peer review file · Nature Communications]

REVIEWER COMMENTS

Reviewer #1 (Remarks to the Author):

The manuscript of Zheng and coworkers present a method to obtain a novel class of chiral gold nanostructure with high dissymmetry factors, up to 15%, through the overgrowth of gold nanodisks in the presence of chiral directing agents glutathione or cysteine. The key finding for controlling the chiral growth relies on the use of bromide and iodide as face-selective capping agents, which facilitates the synthesis of structures with fourfold or threefold rotational symmetry. The effect of the nanodisk size and anisotropy, as well as that of the halide concentration, on the chiral nanocrystal morphology and optical properties, is carefully investigated and characterized. Simulations provide further support for understanding the relationship between the shape and optical features. I find it noticeable that the reported methodology enables the large-scale synthesis of such optically active structures. Overall, the manuscript represents a significant advance in controlling chiral metal nanocrystal growth and I suggest accepting it after minor revisions.

Gold nanoplates are typically multitwinned nanocrystals containing stacking faults, which are partially responsible for the plate morphology (otherwise, it would be challenging to stabilize such highly anisotropic structures). The role of twin defects in metal nanocrystal growth is critical, as certain morphologies can only be obtained when seeds with a given crystal habit are used (for instance, right bipyramids from monotwinned seeds or decahedra from pentatwinned crystals). For this reason, the fact that highly similar chiral structures (432 helicoid III and IV and nanotriskelion) can be obtained from single-crystal (octahedra) and multitwinned seeds (nanodisks derived from plates) is quite surprising. It would be valuable if the authors could provide some insight into this phenomenon. In this sense, I miss in the current manuscript a more detailed discussion about the role of cysteine and glutathione in stabilizing chiral facets. It has been recently proposed that the formation of chiral nanorods with a 422 symmetry arises from the stabilization of $\{521\}$ facets, which are close to the $\{520\}$ found on the original nanorods (Adv. Mater. 2023, 35, 2208299). Perhaps the authors could provide a hypothesis about what facets present in the nanodisks could lead to the emergence of chiral facets during 432 helicoids and nanotriskelion growth or how iodide contributes to the stabilization of such chiral facets.

Reviewer #2 (Remarks to the Author):

The authors presented a halide-assisted differential growth (HADG) strategy that can direct the anisotropic growth of chiral Au nanoparticles. The authors employed Au nanodisks as the seeds to synthesize novel chiral nanoparticles named nanotriskelions. These new nanoparticles have triskelion-shaped chiral surfaces, threefold rotational symmetry, and high dissymmetry factors. Au nanotriskelions were used to realize chiral switching and emission. In the synthesis of chiral nanoparticles, the introduction of KI and CTAB in the growth solution can cause different growth rates along different directions. A series of chiral nanoparticles were therefore obtained, including 432 helicoid III, 432 helicoid IV, and nanotriskelions. It is quite interesting that various chiral nanoparticles can be prepared using the chemical reagents frequently employed in the lab. I recommend its publication, but the following points need to be addressed.

1. Glutathione is the key to the generation of high-Miller-index facets of Au nanoparticles. Have the authors changed the glutathione concentration during the synthesis of Au nanotriskelions?
2. Can the HADG method be used to guide the chiral growth of other Au nanoseeds?
3. The authors used Au nanodisks as the seeds, which were synthesized from triangular Au nanoplates and hexagonal nanoplates. What are the morphologies of the chiral nanoparticles when the triangular and hexagonal nanoplates are used as the seeds?
4. Please provide more information to better show the 3D morphology of the nanotriskelions.

5. Please explain why the Au nanotriskelions deposited on substrates can display a stronger CD response than the nanoparticles in solution.
6. Please provide the units of the electrical field intensity in Fig. 1h and Supplementary Fig. 11.

Reviewer #3 (Remarks to the Author):

The authors present a study on the synthesis of chiral Au nanotriskelions from anisotropic gold nanodisks. They propose a design principle based on halide-assisted differential growth method for creating chirality and detail the growth mechanism. The authors also demonstrate the use of nanotriskelions as optical switching devices and nanoemitters. Their results suggest that the origin of chiral parameters comes from the inorganic metal nanocrystals and offer a methodology for controlling the chirality of nanostructures. The authors have made efforts to provide different chiral nanoparticle morphologies and the study was well executed. The results are suitable for publication in Nature Communications, but certain issues should be addressed before publication.

1. What would happen if the authors adjusted the HADG process to use other nanoplates like triangular and hexagonal nanoplates as seeds instead of Au nanodisks?
2. What is the exact condition of the Au nanotriskelions shown in Fig. 4? The authors present a variety of Au nanotriskelions that are generated from different seeds, Au growth, and halide ions.
3. Why does the peak wavelength difference (~ 200 nm) seem quite different from the results obtained from D-nanoskelions (~ 150 nm) when the substrate for the chiral optical switching device was made by Au L-nanoskelions (Fig. S33d-e)?
4. It is suggested that the authors add more descriptions to justify the need for chiral Au nanoparticles in optical switching devices, as written in lines 321-324.
5. Can the authors provide the result of using chloride ions in the HADG strategy for the synthesis of triskelion nanoparticles instead of bromide and iodide ions?
6. Can the authors provide a detailed explanation for the opposite trend of the g-factor of the nanoparticles in Figure 3c, observed when either L-GSH or D-GSH was applied for the synthesis of triskelion nanoparticles under the presence of either low or high levels of iodide ions?
7. The author discussed the morphology and chirality evolution of Chiral Au nanotriskelions in Figure 3 and demonstrated an opposite chiroptical response (432 helicoid III and Au nanotriskelions) as the dominant symmetry changed from FFR to TFR. However, the structural characterization (FE-SEM images) in the main figure 3 is unclear, and it is better understood from the Supplementary figure 15. A recent paper reported 432 helicoids (Nature 2022, 612, 470-476, Title: Enantioselective sensing by collective circular dichroism) and characterized them, showing four-fold, three-fold, and two-fold rotational symmetry along the $\langle 100 \rangle$, $\langle 111 \rangle$, and $\langle 110 \rangle$ directions. The author should provide a clearer structural characterization.

Reviewer #4 (Remarks to the Author):

Review of the manuscript entitled "Chiral Au nanotriskelions with dominant threefold rotational symmetry"

Manuscript ID: NCOMMS-23-03114-T

Manuscript Type: Article

Recommendation: Minor Revisions

In this work, the authors have reported on a halide-assisted differential growth (HADG) strategy that can direct the anisotropic growth of chiral gold nanoparticles with tunable optical activities and sizes, and rich morphologies. The authors mainly focused on the synthesis of triskelion-shaped chiral nanoparticles with three fold rotational symmetry employing anisotropic nanodisks as seeds and the further application of nanotriskelions as nanoemitters and as chiral optical switching devices. The

authors have also shed light on the generation of other chiral Au nanoparticles by manipulating the directional growth rate by HADG by changing the halide ion concentration as well as the size and shape of the starting seeds. SEM, TEM, HAADF-STEM, 3D electron tomography, and CD analysis together with DDSCAT have been utilized to provide insights about the chiroptical response associated with various chiral nanostructures as well as the associated mechanism for the synthesis.

The HADG synthesis strategy reported for the growth of chiral nanostructures will provide researchers with an enriching library of chiral plasmonic nanoparticles, and is a novel approach with the synthesis of a new kind of nanoparticles, named 'nanotriskelions'. The work is well-contextualized and all the details required for the synthesis have been included in detail in the Supplementary Information. The authors did a great job in telling the story from the synthesis to the potential applications of synthesized chiral nanostructures and I would highly recommend the article to get published in Nature Communications. However, with the implementation of suggested minor changes, the manuscript could become better.

Comments

The title of the manuscript could be more appealing and contained, provided the manuscript deals with the universal method for the synthesis of chiral nanomaterials with morphologies ranging from 432 helicoid to nanotriskelions. I would suggest rephrasing the title with emphasis on 'synthesis' and 'HADG' (Halide Assisted Differential Growth).

Page 2, Line 32: Scattering dissymmetry factor and the g-factor have been used throughout in the manuscript as a measure of chirality for the fabricated chiral nanoparticles but no definition has been provided as such in the main text. It would be nice to include the equations and the associated physical significance for better understanding for the readers. Also, either the maximum or average dissymmetry factor for nanotriskelions could be reported in place of writing 'factor can be larger than 0.5'.

Page 3, Line 81: Instead of writing 'above-mentioned chemical reagents' name of the chemicals (CTAB and KI) that are varied in this manuscript could be mentioned.

Page 5, Fig 1: (e, f, i), Equal sized boxes for the g-factor plots could be used for uniformity. (i), y-axis label should be more descriptive. It should be labelled as a simulation. (h) Scale for $\log |E|$ is difficult to read because of the uneven markings and scaling. (f) Number of particles from which the average is calculated should be mentioned in the main text as well since it is a single particle measurement. Figure 1 is difficult to follow as labelling from (a-i) is not from left to right like other figures in the manuscript. One particular order of labelling the figures should be followed throughout the manuscript.

Page 6, Line 128 and 135: Gold nanotriskelions with highest aspect ratio and largest chiroptical response should be explicitly highlighted with their extinction g-factors mentioned to make the significance of the research more obvious. Instead of writing 'g-factors being higher than 0.1' g-factor obtained should be stated to make the discussion more quantitative and the research work more impactful.

Page 6, Line 142: It is mentioned that the high-quality mass-production synthesis of gold nanotriskelions has been achieved, but no quantitative measure of particle number synthesized or yield has been reported. SEM/TEM images can be used to provide readers with a rough estimate of the number of desired nanoparticles in comparison to other shapes, if any, synthesized.

Page 7, Fig 2: (a-c), 'c(90/30-4)' has been used in all the three figures but with different colour legends. It could be depicted by the same colour to avoid confusion among the figures.

Page 8, Line 169: It should be clearly mentioned in the main text that the seeds employed for the synthesis of 432 helicoid III were 'nanooctahedrons' whereas the 'nanodisks' were used for nanotriskelions.

Page 10, Fig 3: Colour scheme in (a,b) should be changed to make the comparison more distinguishable. Name of the chiral nanostructure along with the structure could be mentioned in (a) to aid the comparison.

Page 10, Line 217-219: The authors have mentioned that "The 432 helicoid nanocrystals are formed at low..." Specific concentrations where Au nanotriskelions start to form should be mentioned.

Page 11, Line 230: The authors have claimed that nanoparticles with opposite chirality must be grown by using ligand enantiomers with opposite handedness. Proper references should be added to support

the statement in the manuscript. One such reference is, Lee, H. E. et. al. Nature Communications 2020, 11.

Page 11, Line 236-238: Diameter-to-thickness ratio should be mentioned along with the respective supplementary figures for both cases, (large ratio needed for Au nanotriskelions and smaller one for cubic helicoid nanocrystals). The discussion is only qualitative with the use of words 'large' and 'small'.

Page 11, Line 240: The approximate size of nanooctahedron seeds should be mentioned for the readers to have a better idea of the growth strategy. Since the authors have used nanodisks of different sizes as seeds, the dimensions of nanooctahedron seeds need to be stated.

Page 12, Line 260: Can the authors comment on the reasoning behind the proposed growth mechanism as to why the increased CTAB and KI concentrations block the directional growth along $\langle 111 \rangle$ and $\langle 100 \rangle$ directions, respectively? Besides the growth mechanism studied using SEM, is there any other experimental evidence (or from the literature) to support the above mentioned conclusions?

Page 13, Line 275: Instead of writing 'various nanophotonic devices', specific applications of nanotriskelions nanostructures (optical switching device and nanoemitter in this manuscript) should be mentioned.

Page 13, Line 279: It's been mentioned that the D-nanotriskelion sample was initially coated the on substrate before using nanotriskelions as the optical chirality device. When Au nanotriskelions are randomly and densely coated on a substrate, what effects can the density and how the nanotriskelions land on the substrate have on their usage as nanophotonic devices? Also, can the authors comment on the usage of different enantiomers of nanotriskelions as nanophotonic devices?

Page 14, Line 295: Acronym (PL) should be defined before its use in the manuscript.

Page 14, Line 312: Rhodamine 640 has been employed in the formation of hybrid nanostructures. Why was R640 chosen? The authors should also briefly mention the thickness of the silica shell of the hybrid nanostructures.

Supplementary Information

Page 15, Line 356: Instead of writing as 'different chiral molecules' 'Cysteine and GSH' could be mentioned.

Page 16, Line 374: Besides plotting the g-factor in Supplementary Fig 9(c, d), a histogram for L- and D- type nanotriskelions should be added together with the calculation of the standard deviation for the g-factor to highlight the structural diversity of the nanoparticles synthesized and the resulting optical properties.

Supplementary Fig 11: The scale bar for (c,d) should be changed to make the plots with differences in the field more visible. With the present colour scheme, it is difficult to draw conclusions from this figure.

Supplementary Fig 31: Can the authors comment on the density of the nanoparticles in solution and the number of nanoparticles on the substrate when they are comparing the g-factor for both the cases?

Reviewer #5 (Remarks to the Author):

The authors studied enhanced chiral gold nanoparticles using a halide-assisted differential growth strategy. The authors studied chirality under various conditions using Au nanodisk, and it is particularly impactful in that the scattering dissymmetry factor is greater than 0.5. However, there are several things to be addressed to be published in Nat. Comm. as below:

1)The greatest novelty of this paper is that the scattering g-factor is greater than 0.5. CDS (Circular Differential Scattering) measurements are said to have been performed on individual Au nanotriskelions lying on substrate, but this does not appear to be a common method. Therefore, it is necessary to perform experiment with the chiral particles made of other 432 helicoid in the same way. In addition, if only chiral nanoparticles made of Au nanodisks have a high scattering g-factor, additional experiments and explanations are needed as to why such a large difference exists.

2)The stability of chiral gold nanoparticles is also important. Experiments on the stability of the created chiral nanoparticles should be necessary.

3)It was very interesting that the 432 helicoid III and Au nanotriskelion had opposite chiroptical responses. However, in the paper, it was inferred only by the different structural characteristics of the two, which is too general answer. It would be better if additional experiments were supported to reveal the mechanism.

Text coding: *black italic, reviewers' comments*; black normal, authors' response; **purple normal, changes made in the manuscript**

Response to Reviewer #1

Comments: *The manuscript of Zheng and coworkers present a method to obtain a novel class of chiral gold nanostructure with high dissymmetry factors, up to 15%, through the overgrowth of gold nanodisks in the presence of chiral directing agents glutathione or cysteine. The key finding for controlling the chiral growth relies on the use of bromide and iodide as face-selective capping agents, which facilitates the synthesis of structures with fourfold or threefold rotational symmetry. The effect of the nanodisk size and anisotropy, as well as that of the halide concentration, on the chiral nanocrystal morphology and optical properties, is carefully investigated and characterized. Simulations provide further support for understanding the relationship between the shape and optical features. I find it noticeable that the reported methodology enables the large-scale synthesis of such optically active structures. Overall, the manuscript represents a significant advance in controlling chiral metal nanocrystal growth and I suggest accepting it after minor revisions.*

Response: We thank this reviewer for the effort on evaluating our work, the highly positive comment, and the insightful questions and suggestions.

Question 1: *Gold nanoplates are typically multitwinned nanocrystals containing stacking faults, which are partially responsible for the plate morphology (otherwise, it would be challenging to stabilize such highly anisotropic structures). The role of twin defects in metal nanocrystal growth is critical, as certain morphologies can only be obtained when seeds with a given crystal habit are used (for instance, right bipyramids from monotwinned seeds or decahedra from pentatwinned crystals). For this reason, the fact that highly similar chiral structures (432 helicoid III and IV and nanotriskelion) can be obtained from single-crystal (octahedra) and multitwinned seeds (nanodisks derived from plates) is quite surprising. It would be valuable if the authors could provide some insight into this phenomenon.*

Response: We thank this reviewer for the insightful suggestion. Previous works have proved that the vertical growth of Ag nanoplates can cause the generation of twinned cubes with truncated corners (*Langmuir* 2014, 30, 15520–15530). We conjecture that 432 helicoid III or IV nanoparticles (NPs) have either twinned or single-crystalline structures. The 432 helicoid III or IV NPs grown from the Au nanodisks have twinned structures, while single-crystalline 432 helicoid NPs are grown from Au octahedrons. We are currently limited by our experimental conditions. However, we hope to perform relevant experiments to verify our conjecture and understand the growth mechanism of Au chiral nanoparticles with twinned structures.

We have added the discussion about this point in the middle of the second paragraph after Figure 2.

“..... Considering the twinned structures of the Au nanodisks, we conjecture that the 432 helicoid III and IV nanoparticles grown from the nanodisks have twinned structures²⁶, compared with the single-crystalline 432 helicoid nanoparticles grown from Au octahedrons³.

The reference (*Langmuir* 2014, 30, 15520–15530) has been added.

26. Zhang, J. W., Liu, J. Y., Xie, Z.-X. & Qin, D. HAuCl₄: A dual agent for studying the chloride-assisted vertical growth of citrate-free Ag nanoplates with Au serving as a marker. *Langmuir* 30, 15520–15530 (2014).

Question 2: *In this sense, I miss in the current manuscript a more detailed discussion about the role of cysteine and glutathione in stabilizing chiral facets.*

Response: Compared with cysteine, glutathione (GSH) is composed of three amino acids and has a longer carbon chain. Previous works have demonstrated that GSH with a larger molecular weight can interact with more kink atoms while cysteine can interact with only a single kink (*Nature* 2018, 556, 360–365). When the growth of the nanotriskelions is guided by GSH, larger protrusions can be produced from the nanodisks, leading to the NPs with an improved optical chirality.

We have added the discussion about this point at the end of the paragraph right before Figure 1.

“..... GSH enantiomers are the chiral directors for the growth of the chiral nanocrystals with opposite handedness (Supplementary Fig. 7). Similar chiral structures can also be produced when GSH is replaced by cysteine (Supplementary Fig. 9). Compared with cysteine, GSH with a larger molecular weight can interact with more kink atoms³. When the growth of the nanotriskelions was guided by GSH, larger protrusions can be produced from the nanodisks, leading to the nanotriskelions with an improved optical chirality.”

Question 3: *It has been recently proposed that the formation of chiral nanorods with a 422 symmetry arises from the stabilization of {521} facets, which are close to the {520} found on the original nanorods (Adv. Mater. 2023, 35, 2208299). Perhaps the authors could provide a hypothesis about what facets present in the nanodisks could lead to the emergence of chiral facets during 432 helicoids and nanotriskelion growth or how iodide contributes to the stabilization of such chiral facets.*

Response: The reducing agent, ascorbic acid, and the chiral ligand, glutathione, are the keys to the generation of high-Miller-index facets and the tilting of the high-Miller-index edges, respectively. Supplementary Fig. 7a shows the SEM image of the NPs synthesized from the growth of the 100/30 nm Au nanodisks without the addition of GSH or KI. The resultant nanocrystals show a hexagram shape and high-Miller-index facets are generated from the {111} facets. The angle between the two salient edges of the Au hexagram-shaped nanocrystals is ~85°, and the angle of the concave edges is ~146°. We hypothesize that the

hexagram-shaped nanocrystals are enclosed by the $\{541\}$ facets according to the literature (*Nano Res.* 2011, 4, 612–622). It still requires high-resolution TEM characterization or theoretical calculation to prove the chiral facets and the contribution of iodide to the stabilization of such chiral facets.

We have revised Supplementary Fig. 7a and added the discussion about this point in the caption of Supplementary Fig. 7a.

Supplementary Fig. 7 | Overgrowth on the Au nanodisks under different conditions. a–c, Overgrowth on the 100/30 nm Au nanodisks without the addition of GSH or KI. The resultant nanocrystals show a hexagram shape and high-Miller-index facets are generated from the $\{111\}$ facets, as revealed by SEM imaging (a). The angle between the two salient edges of the Au hexagram-shaped crystals is $\sim 85^\circ$, and the angle of the concave edges is $\sim 146^\circ$. We hypothesized that the hexagram-shaped nanocrystals are enclosed by the $\{541\}$ facets¹⁵. The HAADF-STEM images (b) and SAED pattern (c) recorded along the $\langle 111 \rangle$ directions reveal that the hexagram-shaped nanocrystals have stacking faults, resulting from the nanodisk seeds¹⁶. The lattice spacing of 0.25 nm is triple that of the $\{422\}$ lattice spacing of the Au nanocrystals, showing the appearance of the diffraction of the $1/3\{422\}$ planes that are normally forbidden by a face-centered cubic lattice. d–f, SEM images showing

We have added the discussion about this point in the paragraph right before Figure 1.

“The Au nanotriskelions with an excellent helicoidal morphology were synthesized in the presence of Au nanodisks, GSH, AA, CTAB, and KI. We then studied the role of the reagents on the growth of the nanotriskelions. The helicoidal morphology of the Au nanotriskelions comes from the generation of high-Miller-index facets on the Au nanodisks. Previous works have demonstrated that the formation of chiral nanorods with a 422 symmetry arises from the stabilization of the $\{521\}$ facets¹⁹. The reducing agent AA is the key to the generation of high-Miller-index facets on the Au nanodisks such as the $\{541\}$ facets (Supplementary Fig. 7)²⁰.

The references (*Adv. Mater.* 2023, 35, 2208299; *Nano Res.* 2011, 4, 612–622) have been added.

19. Ni, B. et al. Chiral seeded growth of gold nanorods into fourfold twisted nanoparticles with plasmonic optical activity. *Adv. Mater.* **35**, 2208299 (2023).

20. Jiang, Q. N. et al. Synthesis and high electrocatalytic performance of hexagram shaped gold particles having an open surface structure with kinks. *Nano Res.* **4**, 612–622 (2011).

Response to Reviewer #2

Comments: The authors presented a halide-assisted differential growth (HADG) strategy that can direct the anisotropic growth of chiral Au nanoparticles. The authors employed Au nanodisks as the seeds to synthesize novel chiral nanoparticles named nanotriskelions. These new nanoparticles have triskelion-shaped chiral surfaces, threefold rotational symmetry, and high dissymmetry factors. Au nanotriskelions were used to realize chiral switching and emission. In the synthesis of chiral nanoparticles, the introduction of KI and CTAB in the growth solution can cause different growth rates along different directions. A series of chiral nanoparticles were therefore obtained, including 432 helicoid III, 432 helicoid IV, and nanotriskelions. It is quite interesting that various chiral nanoparticles can be prepared using the chemical reagents frequently employed in the lab. I recommend its publication, but the following points need to be addressed.

Response: We thank this reviewer for the effort on evaluating our work, the highly positive comment, and the insightful questions and suggestions.

Question 1: Glutathione is the key to the generation of high-Miller-index facets of Au nanoparticles. Have the authors changed the glutathione concentration during the synthesis of Au nanotriskelions?

Response: We thank the reviewer for this insightful question. We considered the GSH concentration during the synthesis of the Au nanotriskelions. The experimental results have been added as Supplementary Fig. 8a, b. The 90/30 nm Au nanodisks were overgrown in the solution made of CTAB (0.1 M, 0.8 mL), KI (1 mM, 40 μ L), AA (0.1 M, 1 mL), HAuCl₄ (0.01 M, 0.4 mL), and D-GSH (2.75 mM). The resultant mixture had a total volume of 10 mL. GSH is the key to the production of the chiral morphology. With the increase in the concentration of D-GSH, the twisted morphology remains almost unchanged, and the protrusion gradually elongates. To investigate the influence of CTAB and KI on the morphology of the chiral NPs, we used a constant concentration for L-GSH (2.75 mM, 150 μ L) and D-GSH (2.75 mM, 180 μ L) in the growth solutions.

Supplementary Fig. 8 | Use of different halide ions to control the chiral growth of the Au nanodisks. a,b, SEM images of the nanoparticles grown from CTAB and KI. The growth solution with a total volume of 10 mL was made of the 90/30 nm nanodisks, CTAB (0.1 M,

0.8 mL), KI (1 mM, 40 μ L), AA (0.1 M, 1 mL), HAuCl₄ (0.01 M, 0.4 mL), and D-GSH (2.75 mM). The generation of Au nanotriskelions in the presence of bromide and iodide ions was observed. With the increase in the concentration of D-GSH, the twisted morphology remains almost unchanged, and the protrusion gradually elongates. **c,d**, SEM images of

Question 2: *Can the HADG method be used to guide the chiral growth of other Au nanoseeds?*

Response: The HADG method can also be used to guide the chiral growth on Au nanocubes. We have synthesized Au nanocubes with an edge length of 45 nm. We found that the Au nanocubes can also evolve into 432 helicoid III and IV by the HADG method, as shown in the figure below. The growth solution was made of CTAB (0.1 M, 0.8 mL), KI, AA (0.1 M, 1 mL), HAuCl₄ (0.01 M, 0.4 mL), and L-GSH (2.75 mM, 150 μ L). The resultant mixture had a total volume of 10 mL. We can observe the effect of the KI concentration on the morphology of the obtained chiral nanocrystals. 432 helicoid III nanocrystals are formed at a low KI concentration (1 mM, 10 μ L). Further addition of KI (1 mM, 30 μ L) finally results in the generation of 432 helicoid IV nanocrystals.

Caption | Au chiral nanocrystals synthesized from Au nanocubes. a) SEM images of the Au nanocubes with an edge length of 45 nm. b) SEM image of the synthesized 432 helicoid III nanocrystals. c) SEM image of the synthesized 432 helicoid IV nanocrystals. L-GSH was employed in the growth experiments.

Based on the experimental results, we have added the discussion on the universality of our HADG method in the Discussion part.

“..... A universal method is employed to prepare a library of plasmonic chiral nanocrystals without the requirement of external stimuli or specific complex reagents during the synthetic process. We believe that the developed HADG method can be used to guide the chiral growth on other Au nanoseeds. In nature, homochiral L-amino acids are used to transfer molecular chirality into biogenic crystals³¹, but the underneath mechanism is”

Question 3: *The authors used Au nanodisks as the seeds, which were synthesized from triangular Au nanoplates and hexagonal nanoplates. What are the morphologies of the chiral nanoparticles when the triangular and hexagonal nanoplates are used as the seeds?*

Response: We have synthesized chiral NPs with the triangular and hexagonal nanoplates used as the seeds. The experimental results are summarized in Supplementary Fig. 17. Supplementary Fig. 17a–d show the SEM images of the NPs synthesized from the Au triangular nanoplates with a side length of 140 nm. The growth solution was made of CTAB (0.1 M, 0.8 mL), KI (1 mM), AA (0.1 M, 1 mL), HAuCl₄ (0.01 M, 0.4 mL), and L-GSH (2.75 mM, 150 μL). The resultant mixture had a total volume of 10 mL. The Au triangular nanoplates can evolve into Au nanotriskelions with segmented arms. The increased KI concentration can cause the gradual embedment of the twisted arms into the nanodisk domain. Supplementary Fig. 17e–h show the SEM images of the NPs synthesized from the hexagonal nanoplates with a side length of 90 nm. The growth solution was made of CTAB (0.1 M, 0.8 mL), KI (1 mM), AA (0.1 M, 1 mL), HAuCl₄ (0.01 M, 0.4 mL), and L-GSH (2.75 mM, 150 μL). The Au hexagonal nanoplates can evolve into NPs with a dendritic morphology. We found that the Au triangular nanoplates and hexagonal nanoplates cannot be used to synthesize Au nanotriskelions with an ideal morphology, which is believed to result from their large sizes. We therefore finally employed the Au nanodisks as the seeds.

Supplementary Fig. 17 | Morphology evolution of the chiral nanocrystals grown from other nanoplates by HADG. **a,b**, Schematic and SEM image of Au triangular nanoplates with side lengths of 140 nm and thicknesses of 10 nm. **c,d**, SEM images showing the morphology evolution of the chiral nanoparticles grown from the triangular nanoplates. The growth solution with a total volume of 10 mL was made of CTAB (0.1 M, 0.8 mL), KI, AA (0.1 M, 1 mL), HAuCl₄ (0.01 M, 0.4 mL), L-GSH (2.75 mM, 150 μL), and DI water. The Au triangular nanoplates can evolve into Au nanotriskelions with segmented arms (**c**). The increased KI concentration can cause the gradual embedment of the twisted arms into the nanodisk domain (**d**). **e,f**, Schematic and SEM image of Au hexagonal nanoplates with side lengths of 90 nm and thicknesses of 30 nm. **g,h**, SEM images showing the morphology evolution of the chiral nanoparticles grown from the hexagonal nanoplates. The growth solution with a total volume of 10 mL was made of CTAB (0.1 M, 0.8 mL), KI, AA (0.1 M, 1 mL), HAuCl₄ (0.01 M, 0.4 mL), L-GSH (2.75 mM, 150 μL), and DI water. The Au hexagonal nanoplates can evolve into the nanoparticles with a dendritic morphology. We found that the Au triangular nanoplates and hexagonal nanoplates cannot be used to synthesize Au nanotriskelions with an ideal morphology, which is believed to be caused by their large aspect ratios. We finally employed the Au nanodisks as the seeds.

We have added the discussion about this point in the paragraph right before Figure 2.

“..... We note that the Au nanodisks with diameter-to-thickness aspect ratios ranging from 1.5 to 4.5 are the key to triggering the anisotropic structure of the nanotriskelions. Other seeds with larger aspect ratios, such as Au triangular and hexagonal nanoplates, can evolve into nanoparticles with dendritic morphologies (Supplementary Fig. 17).”

Question 4: Please provide more information to better show the 3D morphology of the nanotriskelions.

Response: We have provided supplementary movies to visualize the 3D morphologies of the Au nanotriskelions.

Description of additional supplementary files

Supplementary Movie 1 | Tomography reconstruction (left) and constructed model (right) of the L-nanotriskelions.

Supplementary Movie 2 | Tomography reconstruction (left) and constructed model (right) of the D-nanotriskelions.

Question 5: Please explain why the Au nanotriskelions deposited on substrates can display a stronger CD response than the nanoparticles in solution.

Response: We thank this reviewer for the valuable suggestion. We have counted the number of the Au nanotriskelions deposited on the substrates with different orientations. The results show that over 85% of the Au nanotriskelions lie on the substrates. Compared with the randomly orientated Au nanotriskelions in solution, the Au nanotriskelions deposited on the substrates can therefore display a stronger CD response.

We have added the discussion about this point in the paragraph right before Figure 4.

“..... When the Au nanotriskelions are randomly and densely coated on a substrate, we found that over 85% of the Au nanotriskelions can lie on the substrate with their chiral surface facing the incident light. The anisotropic geometric morphology and”

We have also added the discussion about this point in the caption of Supplementary Fig. 35.

“**Supplementary Fig. 35** | **Gold nanotriskelion arrays on substrates.** **a**, Schematic showing the chiroptical response The insets show the zoomed-in SEM images. **Over 85% of the Au nanotriskelions face upward when deposited on the substrates.** **d**, Comparison of”

Question 6: Please provide the units of the electrical field intensity in Fig. 1h and Supplementary Fig. 11.

Response: We have found that the label $\log|\mathbf{E}|$ in the legend of Fig. 1h is incorrect. We think it is better to use the electric field enhancement $|\mathbf{E}|/|\mathbf{E}_0|$ in the legend, where \mathbf{E}_0 is the incident electrical field. In this case, no unit is needed. We have changed the legends, revised the figure caption, and replotted the distributions with different color scales in the revised Fig. 1h and Supplementary Fig. 12.

Response to Reviewer #3

Comments: The authors present a study on the synthesis of chiral Au nanotriskelions from anisotropic gold nanodisks. They propose a design principle based on halide-assisted differential growth method for creating chirality and detail the growth mechanism. The authors also demonstrate the use of nanotriskelions as optical switching devices and nanoemitters. Their results suggest that the origin of chiral parameters comes from the inorganic metal nanocrystals and offer a methodology for controlling the chirality of nanostructures. The authors have made efforts to provide different chiral nanoparticle morphologies and the study was well executed. The results are suitable for publication in Nature Communications, but certain issues should be addressed before publication.

Response: We thank this reviewer for the effort on evaluating our work, the highly positive comment, and the insightful questions and suggestions.

Question 1: What would happen if the authors adjusted the HADG process to use other nanoplates like triangular and hexagonal nanoplates as seeds instead of Au nanodisks?

Response: We thank this reviewer for the valuable question. We have performed the syntheses using the triangular and hexagonal nanoplates as the seeds. The experimental results are summarized in Supplementary Fig. 17. Supplementary Fig. 17a–d show the SEM images of the NPs grown from the Au triangular nanoplates with a side length of 140 nm. The growth solution was made of CTAB (0.1 M, 0.8 mL), KI (1 mM), AA (0.1 M, 1 mL), HAuCl₄ (0.01 M, 0.4 mL), and L-GSH (2.75 mM, 150 μ L). The resultant mixture had a total volume of 10 mL. The Au triangular nanoplates can evolve into Au nanotriskelions with segmented arms. KI at the increased concentration can cause the gradual embedment of the twisted arms into the nanodisk domain. Supplementary Fig. 17e–h show the SEM images of the NPs grown from the hexagonal nanoplates with a side length of 90 nm. The growth solution was made of CTAB (0.1 M, 0.8 mL), KI (1 mM), AA (0.1 M, 1 mL), HAuCl₄ (0.01 M, 0.4 mL), and L-GSH (2.75 mM, 150 μ L). The Au hexagonal nanoplates can evolve into NPs with a dendritic morphology. We found that the Au triangular nanoplates and hexagonal nanoplates cannot be used to synthesize Au nanotriskelions with the ideal morphology, probably because of their large sizes. We therefore finally employed the Au nanodisks as the seeds.

Supplementary Fig. 17 | Morphology evolution of the chiral nanocrystals grown from other nanoplates by HADG. a,b, Schematic and SEM image of Au triangular nanoplates with side lengths of 140 nm and thicknesses of 10 nm. **c,d**, SEM images showing the morphology evolution of the chiral nanoparticles grown from the triangular nanoplates. The growth solution with a total volume of 10 mL was made of CTAB (0.1 M, 0.8 mL), KI, AA (0.1 M, 1 mL), HAuCl₄ (0.01 M, 0.4 mL), L-GSH (2.75 mM, 150 μL), and DI water. The Au triangular nanoplates can evolve into Au nanotriskelions with segmented arms (**c**). The increased KI concentration can cause the gradual embedment of the twisted arms into the nanodisk domain (**d**). **e,f**, Schematic and SEM image of Au hexagonal nanoplates with side lengths of 90 nm and thicknesses of 30 nm. **g,h**, SEM images showing the morphology evolution of the chiral nanoparticles grown from the hexagonal nanoplates. The growth solution with a total volume of 10 mL was made of CTAB (0.1 M, 0.8 mL), KI, AA (0.1 M, 1 mL), HAuCl₄ (0.01 M, 0.4 mL), L-GSH (2.75 mM, 150 μL), and DI water. The Au hexagonal nanoplates can evolve into the nanoparticles with a dendritic morphology. We found that the Au triangular nanoplates and hexagonal nanoplates cannot be used to synthesize Au nanotriskelions with an ideal morphology, which is believed to be caused by their large aspect ratios. We finally employed the Au nanodisks as the seeds.

We have added the discussion about this point in the paragraph right before Figure 2.

“..... We note that the Au nanodisks with diameter-to-thickness aspect ratios ranging from 1.5 to 4.5 are the key to triggering the anisotropic structure of the nanotriskelions. Other seeds with larger aspect ratios, such as Au triangular and hexagonal nanoplates, can evolve into nanoparticles with dendritic morphologies (Supplementary Fig. 17).”

Question 2: *What is the exact condition of the Au nanotriskelions shown in Fig. 4? The authors present a variety of Au nanotriskelions that are generated from different seeds, Au growth, and halide ions.*

Response: We have added the synthesis conditions of the Au nanotriskelions in Supplementary Table 4.

Supplementary Table 4 | Growth conditions of the Au nanotriskelions in Figs. 3, 4, Supplementary Figs. 9, 18, 21, 23, 25, 33, and 34, including the seeds, chiral ligands, and used CTAB/KI concentrations.

Related figure	Seeds	Ligand	CTAB (mM)	KI (μ M)
Fig. 3a, b	90/30	L-GSH	8	0.5, 1.0, 1.2, 1.5, and 2.5
Fig. 3a, b	90/30	D-GSH	8	0.5, 1.0, 1.2, 1.5, and 2.0
Fig. 4b	90/30	D-GSH	12	2.5
Fig. 4e	90/30	L-GSH	8	2.5
Fig. 4e	90/30	D-GSH	8	2.0
Supplementary Fig. 9	90/30	L-Cys	8	3.0
Supplementary Fig. 9	90/30	D-Cys	8	3.0
Supplementary Fig. 18a	90/30	L-GSH	8	2.0
Supplementary Fig. 18b	90/30	L-GSH	8	1.8
Supplementary Fig. 18d	90/30	D-GSH	8	2.5
Supplementary Fig. 18e	90/30	D-GSH	8	2.0
Supplementary Fig. 21a, b	90/45	L-GSH	12	1.2, 1.5, 1.8, 3.0, and 4.0
Supplementary Fig. 21a, b	90/45	D-GSH	12	1.2, 1.5, 2.0, 3.0, and 4.0
Supplementary Fig. 23a, b	90/60	L-GSH	12	1.5, 2.5, 3.0, 4.0, and 5.0
Supplementary Fig. 23a, b	90/60	D-GSH	12	1.5, 2.0, 2.5, 5.0, and 8.0
Supplementary Fig. 25a, b	100/30	L-GSH	8	1.5, 1.8, 2.0, 2.5, and 8.0

Supplementary Fig. 25a, b	100/30	D-GSH	8	1.2, 1.5, 1.8, 3.0, and 8.0
Supplementary Fig. 33a–d	100/30	L-GSH	8	1.8, 2.0, 2.5, and 8.0
Supplementary Fig. 33a–d	100/30	D-GSH	8	1.5, 1.8, 3.0, and 8.0
Supplementary Fig. 33e, f	100/30	L-GSH	6 and 16	2.0
Supplementary Fig. 33e, f	100/30	D-GSH	6 and 16	2.0
Supplementary Fig. 33g–i	100/30	L-GSH	8, 12, and 16	1.5
Supplementary Fig. 33g–i	100/30	D-GSH	8, 12, and 16	1.2
Supplementary Fig. 33j–l	100/30	L-GSH	2, 4, and 6	10
Supplementary Fig. 33j–l	100/30	D-GSH	2, 4, and 6	10
Supplementary Fig. 34b–g	100/30	L-GSH	2, 4, 6, 8, 12, and 16	10
Supplementary Fig. 34h–m	100/30	D-GSH	2, 4, 6, 8, 12, and 16	10

Question 3: Why does the peak wavelength difference (~200 nm) seem quite different from the results obtained from D-nanotriskelions (~150 nm) when the substrate for the chiral optical switching device was made by Au L-nanotriskelions (Fig. S33d-e)?

Response: The distinct peak wavelength differences are caused by the different thicknesses of the PANI layer. Figure 4b shows the D-nanotriskelion-deposited substrate coated with a 60 nm PANI layer. The active chiroptical substrate exhibits a reconfigurable extinction CD peak shift over 100 nm when the coated polyaniline is switched during its proton-doped and dedoped states. Supplementary Fig. 33d–e show the L-nanotriskelion-deposited substrate coated with a 130 nm PANI layer. Peak shifts as large as 200 nm are achieved in the active substrate because of the PANI layer with a larger thickness.

We have added more descriptions to clarify the large peak wavelength difference at the end of the paragraph right before Figure 4.

“..... The obtained active chiral optical substrate exhibited a large and reconfigurable extinction CD peak shift over 100 nm when the coated polyaniline was switched during its proton-doped and dedoped states. Peak shifts as large as 200 nm were also achieved in the active chiral substrate made by the L-nanotriskelion-deposited substrate coated with a 130 nm PANI layer (Supplementary Fig. 37). The modulation of the chiroptical response with excellent stability and reversibility can be potentially applied in switchable polarizers.”

Question 4: *It is suggested that the authors add more descriptions to justify the need for chiral Au nanoparticles in optical switching devices, as written in lines 321-324.*

Response: We have added more descriptions to justify the need for chiral Au nanoparticles in optical switching devices at the end of the paragraph right before Figure 4.

“..... The obtained active chiral optical substrate exhibited a large and reconfigurable extinction CD peak shift over 100 nm when the coated polyaniline was switched during its proton-doped and dedoped states. Peak shifts as large as 200 nm were also achieved in the active chiral substrate made by the L-nanotriskelion-deposited substrate coated with a 130 nm PANI layer (Supplementary Fig. 37). The modulation of the chiroptical response with excellent stability and reversibility can be potentially applied in switchable polarizers.”

Question 5: *Can the authors provide the result of using chloride ions in the HADG strategy for the synthesis of triskelion nanoparticles instead of bromide and iodide ions?*

Response: Yes. We had considered using chloride ions in addition to bromide and iodide ions during the synthesis of the Au nanotriskelions. The experimental results are summarized in Supplementary Fig. 8. Supplementary Fig. 8a and b show the SEM images of the NPs grown from CTAB and KI. The growth solution with a total volume of 10 mL was made of the 90/30 nm nanodisks, CTAB (0.1 M, 0.8 mL), KI (1 mM, 40 μ L), AA (0.1 M, 1 mL), HAuCl₄ (0.01 M, 0.4 mL), and D-GSH (2.75 mM). We can observe the generation of Au nanotriskelions in the presence of bromide and iodide ions. Supplementary Fig. 8c and d show the SEM images of the NPs grown from cetyltrimethylammonium chloride (CTAC). The growth solution with a total volume of 10 mL was made of the 90/30 nm nanodisks, CTAC (0.1 M, 0.8 mL), AA (0.1 M, 1 mL), HAuCl₄ (0.01 M, 0.4 mL), and D-GSH (2.75 mM). We observed that a dendritic morphology with various high-index facets was generated from the nanodisks. The dendritic structures with higher complexity were observed when a larger amount of GSH was used. Previous studies have demonstrated that chloride ions show lower binding affinities on the Au surface than bromide and iodide ions (*J. Am. Chem. Soc.* 2012, 134, 14542–14554). We can conclude that chloride ions cannot be used to control the chiral growth on Au nanodisks.

Supplementary Fig. 8 | Use of different halide ions to control the chiral growth of the Au nanodisks. **a,b**, SEM images of the nanoparticles grown from CTAB and KI. The growth solution with a total volume of 10 mL was made of the 90/30 nm nanodisks, CTAB (0.1 M, 0.8 mL), KI (1 mM, 40 μ L), AA (0.1 M, 1 mL), HAuCl₄ (0.01 M, 0.4 mL), and D-GSH (2.75 mM). The generation of Au nanotriskelions in the presence of bromide and iodide ions was observed. With the increase in the concentration of D-GSH, the twisted morphology remains almost unchanged, and the protrusion gradually elongates. **c,d**, SEM images of the nanoparticles grown from CTAC. The growth solution with a total volume of 10 mL was made of the 90/30 nm nanodisks, CTAC (0.1 M, 0.8 mL), AA (0.1 M, 1 mL), HAuCl₄ (0.01 M, 0.4 mL), and D-GSH (2.75 mM). The dendritic morphology with various high-index facets was generated from the nanodisks (**c**). The dendrite structures with higher complexity were observed when a larger amount of GSH was used (**d**). We therefore conclude that chloride ions cannot be used to control the chiral growth of the Au nanodisks.

We have added more descriptions to show the result of using chloride ions in the HADG strategy in the paragraph right before Figure 1.

“..... We note that chloride ions cannot be used to control the chiral growth of the Au nanodisks because chloride ions show lower binding affinities on the Au surface than bromide and iodide ions (Supplementary Fig. 8)²².”

The reference (*J. Am. Chem. Soc.* 2012, 134, 14542–14554) has been added.

25. Kim, R. M. et al. Enantioselective sensing by collective circular dichroism. *Nature* **612**, 470–476 (2022).

Question 6: *Can the authors provide a detailed explanation for the opposite trend of the g-factor of the nanoparticles in Figure 3c, observed when either L-GSH or D-GSH was applied for the synthesis of triskelion nanoparticles under the presence of either low or high levels of iodide ions?*

Response: We thank the reviewer for this great suggestion. We have added a detailed explanation for the opposite trend of the g-factor in the second paragraph after Figure 2.

“..... When the growth was guided by L-GSH, the 432 helicoid III nanocrystals and the Au nanotriskelions show negative and positive chiroptical response at 500–700 nm, respectively. The 90/30 nm Au nanodisks were found to successively evolve into 432 helicoid III, 432 helicoid IV, and nanotriskelions with increasing KI concentrations (Fig. 3a, also see Supplementary Figs. 3–5). The extinction CD activity is transformed from negative to positive. When the growth was guided by D-GSH, we observed the opposite trend of the chirality evolution with increasing KI concentrations (Fig. 3b).

Question 7: *The author discussed the morphology and chirality evolution of Chiral Au nanotriskelions in Figure 3 and demonstrated an opposite chiroptical response (432 helicoid III and Au nanotriskelions) as the dominant symmetry changed from FFR to TFR. However, the structural characterization (FE-SEM images) in the main figure 3 is unclear, and it is better understood from the Supplementary figure 15. A recent paper reported 432 helicoids (Nature 2022, 612, 470-476, Title: Enantioselective sensing by collective circular dichroism) and characterized them, showing four-fold, three-fold, and two-fold rotational symmetry along the <100>, <111>, and <110> directions. The author should provide a clearer structural characterization.*

Response: We thank the reviewer for this valuable suggestion. The chiral nanocrystals as shown in Fig. 3a oriented along the <100> and <111> directions are summarized in Supplementary Fig. 19. We can clearly observe the dominant symmetry changed from FFR to TFR with the increased KI concentrations.

Supplementary Fig. 19 | Different chiral nanocrystals. a, Synthesized in the presence of L-GSH. **b,** Synthesized in the presence of D-GSH. The SEM images show the chiral nanocrystals in Fig. 3a with the orientation along the $\langle 100 \rangle$ and $\langle 111 \rangle$ directions. The 90/30 nm Au nanodisks were found to successively evolve into 432 helicoid III, 432 helicoid IV, and nanotriskelions with increasing KI concentrations. We can see that the dominant symmetry changes from FFR to TFR with increasing KI concentrations. The scale bars in all the SEM images are the same.

We have also added the relevant description in the second paragraph after Figure 2.

“..... The chiral nanocrystals oriented along the $\langle 100 \rangle$ and $\langle 111 \rangle$ directions are summarized in Supplementary Figure S19. We can see that the dominant symmetry changes from FFR to TFR with increasing KI concentrations. The correlograms on

Response to Reviewer #4

Review of the manuscript entitled “Chiral Au nanotriskelions with dominant threefold rotational symmetry”

Manuscript ID: NCOMMS-23-03114-T

Manuscript Type: Article

Recommendation: Minor Revisions

Comments: *In this work, the authors have reported on a halide-assisted differential growth (HADG) strategy that can direct the anisotropic growth of chiral gold nanoparticles with tunable optical activities and sizes, and rich morphologies. The authors mainly focused on the synthesis of triskelion-shaped chiral nanoparticles with three fold rotational symmetry employing anisotropic nanodisks as seeds and the further application of nanotriskelions as nanoemitters and as chiral optical switching devices. The authors have also shed light on the generation of other chiral Au nanoparticles by manipulating the directional growth rate by HADG by changing the halide ion concentration as well as the size and shape of the starting seeds. SEM, TEM, HAADF-STEM, 3D electron tomography, and CD analysis together with DDSCAT have been utilized to provide insights about the chiroptical response associated with various chiral nanostructures as well as the associated mechanism for the synthesis.*

The HADG synthesis strategy reported for the growth of chiral nanostructures will provide researchers with an enriching library of chiral plasmonic nanoparticles, and is a novel approach with the synthesis of a new kind of nanoparticles, named ‘nanotriskelions’. The work is well-contextualized and all the details required for the synthesis have been included in detail in the Supplementary Information. The authors did a great job in telling the story from the synthesis to the potential applications of synthesized chiral nanostructures and I would highly recommend the article to get published in Nature Communications. However, with the implementation of suggested minor changes, the manuscript could become better.

Response: We thank this reviewer for the effort on evaluating our work, the highly positive comment, and the insightful questions and suggestions.

Question 1: *The title of the manuscript could be more appealing and contained, provided the manuscript deals with the universal method for the synthesis of chiral nanomaterials with morphologies ranging from 432 helicoid to nanotriskelions. I would suggest rephrasing the title with emphasis on ‘synthesis’ and ‘HADG’ (Halide Assisted Differential Growth).*

Response: We thank the reviewer for this valuable suggestion. We have revised the title to “Halide-assisted differential growth for the synthesis of chiral 432 helicoid nanoparticles and nanotriskelions”.

Question 2: *Page 2, Line 32: Scattering dissymmetry factor and the g-factor have been used throughout in the manuscript as a measure of chirality for the fabricated chiral nanoparticles but no definition has been provided as such in the main text. It would be nice to include the equations and the associated physical significance for better understanding for the readers. Also, either the maximum or average dissymmetry factor for nanotriskelions could be reported in place of writing ‘factor can be larger than 0.5’.*

Response: We thank the reviewer for this valuable suggestion. We have provided the relevant discussion on the equations and physical significance of the extinction g-factor and scattering g-factor. The revision is made in the first paragraph right after Figure 1.

“The helicoidal morphology of the nanotriskelions results in excellent chiroptical properties. The extinction circular dichroism (CD) spectra of the Au nanotriskelions were measured on a CD spectrophotometer. The extinction dissymmetry factor (*g*-factor) was obtained to quantify the chiroptical behavior of the nanocrystals in solution, as described by³

$$g = 2 \times \frac{A_L - A_R}{A_L + A_R} \quad (1)$$

where A_L and A_R are the extinction spectra for the left-handed circularly polarized (LCP) and right-handed circularly polarized (RCP) incident light, respectively. The opposite rotation directions of the arms on

“..... were therefore performed on the individual Au nanotriskelions lying on substrates (Supplementary Fig. 10)^{23,24}. The scattering spectra of the individual nanoparticles were measured under the excitation of LCP and RCP light (S_{LCP} and S_{RCP}). The scattering dissymmetry factor (g_s) was calculated by

$$g_s = 2 \times \frac{S_{LCP} - S_{RCP}}{S_{LCP} + S_{RCP}} \quad (2)$$

We have measured the CDS spectra for a number of nanotriskelions, including 41 L-type and 32 D-type ones.”

We have also revised the description on the average dissymmetry factor for the nanotriskelions in the abstract and in the middle of the first paragraph right after Figure 1.

“..... are named nanotriskelions. The averaged scattering *g*-factors of the L- and D-nanotriskelions are as large as 0.57 and -0.49 at 650 nm, respectively. The Au nanotriskelions have been

“..... The averaged scattering *g*-factors of the L- and D-nanotriskelions are as large as 0.57 and -0.49 at 650 nm, respectively (Fig. 1f). The geometry obtained from

Question 3: Page 3, Line 81: Instead of writing ‘above-mentioned chemical reagents’ name of the chemicals (CTAB and KI) that are varied in this manuscript could be mentioned.

Response: We have made the suggested change.

“..... These nanocrystals were obtained by adjusting the amounts of CTAB and KI that are frequently employed during

Question 4: Page 5, Fig 1: (e, f, i), Equal sized boxes for the *g*-factor plots could be used for uniformity. (i), y-axis label should be more descriptive. It should be labelled as a simulation. (h) Scale for log |*E*| is difficult to read because of the uneven markings and scaling. (f) Number of particles from which the average is calculated should be mentioned in the main text as well since it is a single particle measurement.

Response: The questions are addressed one by one below.

(1) We have used equally sized boxes in Fig. 1c, f, and i.

(2) We have labelled Fig. 1i as the simulation result and labelled Fig. 1f as the experimental result.

(3) We have found that the label $\log|\mathbf{E}|$ in the legend of Fig. 1h is incorrect. We think it is better to use the electric field enhancement $|\mathbf{E}|/|\mathbf{E}_0|$ in the legend, where \mathbf{E}_0 is the incident electrical field. We have changed the legends, revised the figure caption, and replotted the distributions with different color scales in the revised Fig. 1h as well as Supplementary Fig. 12.

Fig. 1 | Morphology and chiroptical properties of the Au nanotriskelions. **a**, SEM images of the Au L-nanotriskelions. **b**, Tomography reconstruction (top) and constructed model (bottom) of the L-nanotriskelions. **c**, Dissymmetry factor spectra measured from CD spectrometry for the L- and D-nanotriskelions in solution. **d**, SEM images of the Au D-nanotriskelions. **e**, Tomography reconstruction (top) and constructed model (bottom) of the D-nanotriskelions. The Au nanotriskelions in **(a,d)** were grown from the 90/30 nm (diameter/thickness) nanodisks in the presence of L- and D-GSH. All the scale bars in **(a,d)** are the same. **f**, Average scattering g -factor spectra obtained from the CDS measurements on the L- and D-nanotriskelions. **g**, DDSCAT simulation of the CDS of the L-nanotriskelions. The scattering intensities excited by the linearly polarized (LP), LCP, and RCP light were calculated. **h**, Electric fields of the L-nanotriskelion under the excitation of LCP and RCP light at 630 nm. The distributions of the electric field enhancement are shown in the color maps at the logarithmic scale. **i**, Simulated scattering g -factor spectra of the typical L- and D-nanotriskelions.

(4) We have also mentioned the number of the nanotriskelions for the CDS measurements in the first paragraph right after Figure 1.

“..... We have measured the CDS spectra for a number of nanotriskelions, including 41 L-type and 32 D-type ones.”

Question 5: *Figure 1 is difficult to follow as labelling from (a-i) is not from left to right like other figures in the manuscript. One particular order of labelling the figures should be followed throughout the manuscript.*

Response: We have revised Fig. 1 and the caption according to the suggestion. Please see the response to Question 4 above for Fig. 1.

Question 6: *Page 6, Line 128 and 135: Gold nanotriskelions with highest aspect ratio and largest chiroptical response should be explicitly highlighted with their extinction g-factors mentioned to make the significance of the research more obvious. Instead of writing ‘g-factors being higher than 0.1’ g-factor obtained should be stated to make the discussion more quantitative and the research work more impactful.*

Response: We thank the reviewer for this valuable suggestion. We have changed the obscure description in the middle of the paragraph right before Figure 2.

“..... We can therefore obtain the Au nanotriskelions with large aspect ratios and strong chiroptical response. For example, the L-Au nanotriskelions grown from the 100/30 nm (diameter/thickness) nanodisks show a high aspect ratio at ~ 2.1 and display enhanced chiroptical response with an extinction g-factor of 0.12 at 616 nm (Supplementary Table 2).”

Question 7: *Page 6, Line 142: It is mentioned that the high-quality mass-production synthesis of gold nanotriskelions has been achieved, but no quantitative measure of particle number synthesized or yield has been reported. SEM/TEM images can be used to provide readers with a rough estimate of the number of desired nanoparticles in comparison to other shapes, if any, synthesized.*

Response: We have counted the numbers of the Au nanotriskelions with desired structures from SEM images. The yield of the Au nanotriskelions was estimated to be larger than 80%, as shown in the revised Supplementary Fig. 18. We have added the description of the NP yield in the paragraph right before Fig. 2.

“..... Additionally, the HADG strategy guarantees the mass-production synthesis of the Au nanotriskelions with a yield larger than 80% (Supplementary Fig. 18a–f). The volume of the growth solution for”

Supplementary Fig. 18 | Scale-up synthesis and structural stability of the Au nanotriskelions. a–c, SEM images and extinction g -factor spectra of the L-nanotriskelions. d–f, SEM images and extinction g -factor spectra of the D-nanotriskelions. The insets in (a, b, d, and e) are the photographs of the nanotriskelion solutions. The HADG synthetic strategy was demonstrated to support high-quality and mass production. The synthesis of the Au nanotriskelions has been scaled up, with the total volume of the growth solutions expanded to 40 mL and 120 mL, respectively. The high quality was also ensured, as demonstrated by the SEM images and extinction g -factor spectra. We counted the number of the Au nanotriskelions with the desired structures from the SEM images. The number yield of the Au nanotriskelions was estimated to be larger than 80%. g, h, SEM images of the D-nanotriskelions and the same sample stored after 6 months. The nanotriskelions were stored in an environment of 0–5 °C. The morphology can be preserved well for at least 6 months.

Question 8: Page 7, Fig 2: (a-c), 'c(90/30-4)' has been used in all the three figures but with different colour legends. It could be depicted by the same colour to avoid confusion among the figures.

Response: We thank the reviewer for carefully reading the manuscript. We have revised Fig. 2 according to the suggestion.

Question 9: Page 8, Line 169: It should be clearly mentioned in the main text that the seeds employed for the synthesis of 432 helicoid III were ‘nanooctahedrons’ whereas the ‘nanodisks’ were used for nanotriskelions.

Response: We have changed the obscure description in the second paragraph after Fig. 1.

“..... Although the 432 helicoid III nanocrystals (grown from octahedrons) and the Au nanotriskelions (grown from nanodisks) are prepared in the presence of GSH with the same handedness, the nanotriskelions viewed from the <111> directions and the cubic helicoid nanocrystals viewed from the <100> directions exhibit opposite geometrical chirality (Supplementary Fig. 13).

Question 10: Page 10, Fig 3: Colour scheme in (a,b) should be changed to make the comparison more distinguishable. Name of the chiral nanostructure along with the structure could be mentioned in (a) to aid the comparison.

Response: We have revised Fig. 3 according to the suggestion.

Question 11: Page 10, Line 217-219: The authors have mentioned that “The 432 helicoid nanocrystals are formed at low...” Specific concentrations where Au nanotriskelions start to form should be mentioned.

Response: We have added the specific concentrations in the caption of Fig. 3.

“..... The 432 helicoid III nanocrystals are formed at low Γ^- concentrations (0.5–1.0 μM) and the 432 helicoid IV nanocrystals start to appear as the Γ^- concentration is increased. Further addition of KI (1.8–8.0 μM) finally results in the generation of the Au nanotriskelions. The morphology evolution was

Question 12: Page 11, Line 230: The authors have claimed that nanoparticles with opposite chirality must be grown by using ligand enantiomers with opposite handedness. Proper references should be added to support the statement in the manuscript. One such reference is, Lee, H. E. et. al. Nature Communications 2020, 11.

Response: We have added the reference citations to support the statement in the paragraph after Fig. 3.

“..... The result is against the perception that nanoparticles with opposite chiroptical response must be grown by use of ligand enantiomers with opposite handedness^{3,7}. Our HADG process may

The two references were cited in our original manuscript.

3. Lee, H.-E. et al. Amino-acid- and peptide-directed synthesis of chiral plasmonic gold nanoparticles. *Nature* **556**, 360–365 (2018).

7. Lee, H.-E. et al. Cysteine-encoded chirality evolution in plasmonic rhombic dodecahedral gold nanoparticles. *Nat. Commun.* **11**, 263 (2020).

Question 13: Page 11, Line 236-238: Diameter-to-thickness ratio should be mentioned along with the respective supplementary figures for both cases, (large ratio needed for Au nanotriskelions and smaller one for cubic helicoid nanocrystals). The discussion is only qualitative with the use of words 'large' and 'small'.

Response: We have added the diameter-to-thickness ratios in the second paragraph after Fig. 3 and in the captions of Fig. 3 and Supplementary Figs. 21, 23, 25.

“..... Such dependence is highly determined by the aspect ratio of the Au nanodisk seeds. The Au nanodisks with the diameter-to-thickness ratio at 3.3 (more anisotropic) prefer to grow into Au nanotriskelions while the Au nanodisks with the diameter-to-thickness ratios at 1.5 (more isotropic) tend to give rise to cubic helicoid nanocrystals. To reveal

“**Fig. 3 | Morphology and chirality evolution of the chiral nanocrystals grown from the Au nanodisks by HADG.** The Au nanodisks with a size of 90/30 nm and diameter-to-thickness ratio of 3.0 were employed as the seeds. **a**, Effect of the KI concentration on

“**Supplementary Fig. 21 | Morphology and chirality evolution of the chiral nanocrystals grown from the Au nanodisks by HADG.** The Au nanodisks with a size of 90/45 nm and diameter-to-thickness ratio of 2.0 were employed as the seeds.....”

“**Supplementary Fig. 23 | Morphology and chirality evolution of the chiral nanocrystals grown from the Au nanodisks by HADG.** The Au nanodisks with a size of 90/60 nm and diameter-to-thickness ratio of 1.5 were employed as the seeds.....”

“**Supplementary Fig. 25 | Morphology and chirality evolution of the chiral nanocrystals grown from the Au nanodisks by HADG.** The Au nanodisks with a size of 100/30 nm and diameter-to-thickness ratio of 3.3 were employed as the seeds.....”

Question 14: Page 11, Line 240: The approximate size of nanooctahedron seeds should be mentioned for the readers to have a better idea of the growth strategy. Since the authors have used nanodisks of different sizes as seeds, the dimensions of nanooctahedron seeds need to be stated.

Response: The octahedrons have the edge length of 55 nm. We have added the SEM images of the octahedrons in Supplementary Fig. 28. We have mentioned the size of the nanooctahedrons in the second paragraph after Fig. 3 and the captions of Supplementary Figs. 28 and 29.

“..... The octahedrons with edge lengths of 55 nm were found to steer the generation of the 432 helicoid nanocrystals.....”

“**Supplementary Fig. 28 | Morphology and chirality evolution of the chiral nanocrystals grown from Au octahedrons by HADG. a, Model and SEM image of the Au octahedrons with edge lengths of 55 nm.**”

“**Supplementary Fig. 29 | Effect of the CTAB concentration on the morphology evolution of the 432 helicoid III nanocrystals grown through HADG on the Au octahedrons with edge lengths of 55 nm.**”

Question 15: Page 12, Line 260: Can the authors comment on the reasoning behind the proposed growth mechanism as to why the increased CTAB and KI concentrations block the directional growth along $\langle 111 \rangle$ and $\langle 100 \rangle$ directions, respectively? Besides the growth mechanism studied using SEM, is there any other experimental evidence (or from the literature) to support the above-mentioned conclusions?

Response: We thank this reviewer for the insightful suggestions. They are addressed one by one below.

(1) It has generally been accepted that halide ions can bind to the Au surface with different binding affinities in the order of $I^- > Br^- > Cl^-$ (*J. Am. Chem. Soc.* **2012**, *134*, 14542–14554). Iodide ions favorably bind to the $\{111\}$ facets and reduce the Au deposition rate on the $\{111\}$ facets. Bromide ions, on the other hand, are preferentially adsorbed on the $\{100\}$ facets. Directional growth along the $\langle 111 \rangle$ and $\langle 100 \rangle$ directions can therefore be blocked by KI and CTAB at increased concentrations, respectively.

(2) In our experiment, we have synthesized various seeds including the octahedrons and nanodisks encapsulated with the $\{111\}$ facets. KI is the essential agent in the synthesis of Au octahedrons and Au nanoplates (*Nat. Commun.* 2019, **10**, 5789). Previous works have demonstrated that bromide ions at increased concentrations can trigger the growth of 10 nm Au NPs into nanocubes encapsulated with the $\{100\}$ facets (*Nano Lett.* **2018**, *18*, 6475–6482). Based on the literature and synthesis procedures, we had confirmed that CTAB and KI at the increased concentrations block the directional growth along the $\langle 111 \rangle$ and $\langle 100 \rangle$ directions.

We have added the description for using CTAB and KI to block the directional growth in the paragraph right before Figure 1.

“..... CTAB and KI are employed as the growth regulators. Iodide ions favorably bind to the $\{111\}$ facets and reduce the Au deposition rate on the $\{111\}$ facets^{13,14}. Bromide ions, on the other hand, are preferentially adsorbed on the $\{100\}$ facets²¹. The directional growth along the $\langle 100 \rangle$ and $\langle 111 \rangle$ directions can be blocked with increased CTAB and KI concentrations, respectively. We note that chloride ions cannot be used to control the chiral growth of the Au nanodisks because chloride ions show lower binding affinities on the Au surface than bromide and iodide ions (Supplementary Fig. 8)²².”

We have cited and added the relevant references to support our discussion.

13. Yoo, S., Kim, J., Choi, S., Park, D. & Park, S. Two-dimensional nanoframes with dual rims. *Nat. Commun.* **10**, 5789 (2019).
14. Haddadnezhad, M. et al. Synthesis and surface plasmonic characterization of asymmetric Au split nanorings. *Nano Lett.* **20**, 7774–7782 (2020).
21. Park, J.-E., Lee, Y. & Nam, J.-M. Precisely shaped, uniformly formed gold nanocubes with ultrahigh reproducibility in single-particle scattering and surface-enhanced Raman scattering. *Nano Lett.* **18**, 6475–6482 (2018).
22. Langille, M. R., Personick, M. L., Zhang, J. & Mirkin, C. A. Defining rules for the shape evolution of gold nanoparticles. *J. Am. Chem. Soc.* **134**, 14542–14554 (2012).

Question 16: Page 13, Line 275: Instead of writing ‘various nanophotonic devices’, specific applications of nanotriskelions nanostructures (optical switching device and nanoemitter in this manuscript) should be mentioned.

Response: We thank the reviewer for the valuable suggestion. We have changed the obscure sentence in the subheading and the caption of Fig. 4.

“Au nanotriskelions for **chiroptical switching and emissions.**”

“**Fig. 4 | Au nanotriskelions as a platform to realize chiroptical switching and emissions.**”

Question 17: Page 13, Line 279: *It’s been mentioned that the D-nanotriskelion sample was initially coated the on substrate before using nanotriskelions as the optical chirality device. When Au nanotriskelions are randomly and densely coated on a substrate, what effects can the density and how the nanotriskelions land on the substrate have on their usage as nanophotonic devices? Also, can the authors comment on the usage of different enantiomers of nanotriskelions as nanophotonic devices?*

Response: The questions are addressed one by one below.

(1) When the chiral NP density is increased, more chiral structures are involved in the absorption and scattering of incident light, which results in improved chiral response. The enhanced chiroptical response was observed with increasing NP number densities.

(2) We believe that the chiroptical response of the chiral NPs can be significantly enhanced when the chiral surfaces are facing toward incident light. The Au nanotriskelions lying on substrates with the chiral surfaces facing upwards can be the best candidate for nanophotonic device applications. When the Au nanotriskelions were deposited on the substrate, we found that over 85% of the Au nanotriskelions were lying on the substrate. We can therefore obtain enhanced chiroptical response.

(3) Assembly of chiral NPs onto a substrate might exhibit strong optical anisotropy and produce special response to incident light. These features enable the resultant chiral plasmonic substrate to be used for spatial light modulators (*Nature* 2022, **612**, 470–476),

enantioselective sensors (*Nano Lett.* 2018, **18**, 6279–6285), and other chiral nanophotonic devices.

We have revised the relevant description in the paragraph right before Fig. 4.

“We further demonstrated the use of the Au nanotriskelions as a platform to realize different nanophotonic functionalities. The assembly of the chiral nanoparticles onto substrates can exhibit strong optical anisotropy and produce special response to incident light. These features enable chiral plasmonic substrates potentially as spatial light modulators²⁵, enantioselective sensors²⁷, and other chiral nanophotonic devices. When the Au nanotriskelions are randomly and densely coated on a substrate, we found that over 85% of the Au nanotriskelions can lie on the substrate with their chiral surface facing the incident light. The anisotropic geometric morphology and chirality enable the Au nanotriskelions deposited on substrates to display a stronger CD response than the nanocrystals in solution (Supplementary Fig. 35). When the chiral nanoparticle density is increased, more chiral structures are involved in the absorption and scattering of the incident light, resulting in the improved chiral response (Supplementary Fig. 36). A chiral optical switching device was

Two references have been added to support our discussion.

25. Kim, R. M. et al. Enantioselective sensing by collective circular dichroism. *Nature* **612**, 470–476 (2022).

27. García-Guirado, J., Svedendahl, M., Puigdollers, J. & Quidant, R. Enantiomer-selective molecular sensing using racemic nanoplasmonic arrays. *Nano Lett.* **18**, 6279–6285 (2018).

Question 18: Page 14, Line 295: Acronym (PL) should be defined before its use in the manuscript.

Response: We have defined the acronym (PL) in the paragraph right after Fig. 4.

“..... Under the excitation of LCP and RCP laser light, the core@shell nanostructures exhibit largely different photoluminescence (PL) intensities, suggesting that the chiral nanocrystals remarkably modify

Question 19: Page 14, Line 312: Rhodamine 640 has been employed in the formation of hybrid nanostructures. Why was R640 chosen? The authors should also briefly mention the thickness of the silica shell of the hybrid nanostructures.

Response: Rhodamine 640 (R640), showing strong absorption at 575 nm and emission at 620–650 nm, was selected as the organic fluorophore for chiral emission (*Laser Phys.* 2015, **25**, 085001). The thickness of the silica shell was 10–15 nm.

We have added the reason for using Rhodamine 640 as the fluorophore and the thickness of the silica shell in the paragraph right after Fig. 4.

“In addition to the chiroptical switching device, we also demonstrated the feasibility of constructing nanoemitters that preferentially emit LCP or RCP photons from the individual Au nanotriskelions. Rhodamine 640 (R640), showing strong absorption at 575 nm and emission at 620–650 nm, was selected as the organic fluorophore for chiral emission³⁰. (Gold nanotriskelion)–fluorophore hybrid nanostructures were prepared by embedding Rhodamine 640 (R640) into the shell of the silica-coated nanotriskelions (Fig. 4d–j and Supplementary Fig. 38). The thickness of the silica shell was 10–15 nm. The Au nanotriskelions can largely scatter and absorb CP light at 590–650 nm. Under the excitation at 594 nm, the interaction between the excitons in R640 and the plasmonic chiral near-field triggers plasmon–exciton coupling that is sensitive to the polarization of the excitation laser light (Fig. 4f, g). The emitted photons can be scattered by the Au nanotriskelions, resulting in enhanced chiral emissions. Under the excitation of

A reference has been added to support our discussion.

30. Ismail, W. Z. W., Vo, T. P., Goldys, E. M. & Dawes, J. M. Plasmonic enhancement of Rhodamine dye random lasers. *Laser Phys.* **25**, 085001 (2015).

Question 20: *Supplementary Information (Page 15, Line 356): Instead of writing as ‘different chiral molecules’ ‘Cysteine and GSH’ could be mentioned.*

Response: We have changed the unclear description in the caption of Supplementary Fig. 9.

“**Supplementary Fig. 9 | Structural difference of the Au nanotriskelions synthesized in the presence of Cys and GSH.** a,b, SEM images and *g*-factor spectra of the Au nanotriskelions obtained from L- and D-Cys.....”

Question 21: *Page 16, Line 374: Besides plotting the *g*-factor in Supplementary Fig 9(c, d), a histogram for L- and D- type nanotriskelions should be added together with the calculation of the standard deviation for the *g*-factor to highlight the structural diversity of the nanoparticles synthesized and the resulting optical properties.*

Response: We have provided the histograms for the scattering *g*-factors of the L- and D-nanotriskelions and added the statistical results for the scattering *g*-factors in Supplementary Fig. 10.

Supplementary Fig. 10 | CDS measurements of the Au nanotriskelions. a,b, SEM-correlated single-particle CDS spectra for the L- (a) and D-nanotriskelions (b) under LCP and RCP excitation at 550–750 nm..... **e,f,** Histograms for the L- and D-type nanotriskelions with the standard deviations for the scattering g-factor at 650 nm.

Question 22: Supplementary Fig 11: The scale bar for (c,d) should be changed to make the plots with differences in the field more visible. With the present colour scheme, it is difficult to draw conclusions from this figure.

Response: We have changed the legends and replotted the distributions with different color scales in the revised Supplementary Fig. 12.

Question 23: Supplementary Fig 31: Can the authors comment on the density of the nanoparticles in solution and the number of nanoparticles on the substrate when they are comparing the g-factor for both the cases?

Response: We are very sorry for that we cannot provide the specific densities of the nanotriskelions in the solutions and the numbers of the nanotriskelions on the substrates. We

only measured the CD spectra of the same nanotriskelions in the original growth solutions and on the substrates. The absorption intensities in the solutions and on the substrates might be different because the numbers of the Au nanotriskelions might be different. We calculated the *g*-factors to eliminate the influence of the absorption difference. We have added the relevant discussion in the caption of Supplementary Fig. 35 as follows.

“Supplementary Fig. 35 | Gold nanotriskelion arrays on substrates. a, Schematic showing the chiroptical response of the Au nanotriskelion-deposited substrate..... **d,** Comparison of the extinction *g*-factors for the Au nanotriskelions deposited on silica substrates and dispersed in solution. **The CD intensities can be different because of the absorption difference between the solution and the substrate. We therefore calculated the *g*-factors to eliminate the influence of the absorption difference.** The chiroptical response of the colloidal nanocrystals dispersed in solution is weaker than that of the same nanocrystals deposited on the substrates, which results from the random orientation of the nanocrystals in solution.”

Response to Reviewer #5

Comments: *The authors studied enhanced chiral gold nanoparticles using a halide-assisted differential growth strategy. The authors studied chirality under various conditions using Au nanodisk, and it is particularly impactful in that the scattering dissymmetry factor is greater than 0.5. However, there are several things to be addressed to be published in Nat. Comm. as below:*

Response: We thank this reviewer for the effort on evaluating our work, the highly positive comment, and the insightful questions and suggestions.

Question 1: *The greatest novelty of this paper is that the scattering *g*-factor is greater than 0.5. CDS (Circular Differential Scattering) measurements are said to have been performed on individual Au nanotriskelions lying on substrate, but this does not appear to be a common method. Therefore, it is necessary to perform experiment with the chiral particles made of other 432 helicoid in the same way. In addition, if only chiral nanoparticles made of Au nanodisks have a high scattering *g*-factor, additional experiments and explanations are needed as to why such a large difference exists.*

Response: We thank the reviewer for this insightful suggestion. The questions are addressed one by one below.

(1) We have performed the CDS measurements on the individual 432 helicoid III NPs deposited on silica substrates. The experimental results have been added as Supplementary Fig. 14a–c. We have calculated the average scattering *g*-factors for the 432 helicoid III NPs, including 14 L-type and 11 D-type NPs. The L-432 helicoid III NPs exhibit an extinction *g*-factor of 0.088 at 590 nm and the average scattering *g*-factor is 0.45 at 640 nm. The D-432 helicoid III NPs exhibit an extinction *g*-factor of 0.05 at 590 nm and the average scattering *g*-factor is 0.3 at 640 nm. Compared with the experimental results shown in Fig. 1a–f, we can

find that CDS measurements can be an effective method to characterize the chiroptical response of chiral NPs.

(2) To explain why the Au nanotriskelions show the large scattering g -factor, we have measured the CDS spectra of the L-432 helicoid III NPs with different tilting angles, as shown in Supplementary Fig. 14d–f. When the chiral surface faces upwards, the L-432 helicoid III NPs show a stronger scattering peak under RCP excitation. The scattering g -factor was calculated to be 1.0 at 625 nm. On the other hand, when the chiral surface does not face upwards, the L-432 helicoid III NPs show a reduced scattering signal under RCP excitation. The scattering g -factor was calculated to be 0.2 at 625 nm. When the chiral surface faces upwards, the differential scattering signal from the chiral surface can be received more efficiently, leading to the increased scattering g -factor. When the Au nanotriskelions are deposited on substrates, their chiral surfaces always face upwards because of the anisotropic geometric morphology, as shown in Supplementary Fig. S10. As a result, the Au nanotriskelions show a higher scattering g -factor compared with the 432 helicoid III NPs.

The experimental results have been added as Supplementary Figure S14.

Supplementary Fig. 14 | CDS measurements of the 432 helicoid III NPs. **a**, Constructed models and SEM images of the 432 helicoid III NPs. **b**, Dissymmetry factor spectra measured from CD spectrometry for the 432 helicoid III NPs in solution. **c**, Average scattering g -factor spectra obtained from the CDS measurements on the 432 helicoid III NPs, including 14 L-type and 11 D-type NPs. **d,e**, SEM-correlated single-particle CDS spectra for the L-432 helicoid III with different tilted angles under LCP and RCP excitation at 550–750 nm. Insets: SEM images of the measured chiral nanocrystals. The SEM images of the measured nanotriskelions confirm that the differential scattering results from their different tilted angles. **f**, Scattering g -factor spectra for the L-432 helicoid III in **(d,e)**. When the chiral

surface faces upwards, the L-432 helicoid III shows the stronger differential scattering response.

We have also added the relevant discussion in the second paragraph after Fig. 1.

“A combination of CD and CDS measurements is helpful in comprehensively revealing the relation between the helicoidal structure and the chiroptical response of chiral plasmonic nanoparticles. We next studied the differences in structural and chiroptical responses between the 432 helicoid III and Au nanotriskelions (Supplementary Figs. 13, 14). The 432 helicoid III nanocrystals with strong chiroptical properties were synthesized from Au octahedrons and are of a cubic geometry^{3,25}. Their six faces exhibit pinwheel-like patterns that consist of four highly curved arms of increasing widths. The strong chiroptical properties of the 432 helicoid III and Au nanotriskelions stem from their chiral geometries, which dominantly exhibit fourfold rotational (FFR) symmetry along the $\langle 100 \rangle$ directions and TFR symmetry along the $\langle 111 \rangle$ directions, respectively. Although the 432 helicoid III nanocrystals (grown from octahedrons) and the Au nanotriskelions (grown from nanodisks) are prepared in the presence of GSH with the same handedness, the nanotriskelions viewed from the $\langle 111 \rangle$ directions and the cubic helicoid nanocrystals viewed from the $\langle 100 \rangle$ directions exhibit opposite geometrical chirality (Supplementary Fig. 13). As a result, they exhibit opposite chiroptical responses in the wavelength range of 500–700 nm. For example, Supplementary Fig. 14 shows that the L-432 helicoid III exhibits an extinction g -factor of -0.088 at 590 nm and the average scattering g -factor is -0.45 at 640 nm. On the other hand, the L-nanotriskelions (Fig. 1c, f) show an extinction g -factor of 0.05 at 595 nm and the average scattering g -factor is 0.57 at 650 nm.”

Question 2: *The stability of chiral gold nanoparticles is also important. Experiments on the stability of the created chiral nanoparticles should be necessary.*

Response: We have tested the structural stability of the chiral Au NPs. When the Au nanotriskelions were stored in an environment of $0-5\text{ }^{\circ}\text{C}$, we found that the morphology of the nanotriskelions can be preserved well for at least 6 months from SEM imaging. The Au nanotriskelions therefore show good stability. We have added the SEM images as Supplementary Fig. 18g and h.

Supplementary Fig. 18 | Scale-up synthesis and structural stability of the Au nanotriskelions. a–c, SEM images and extinction g -factor spectra of the L-nanotriskelions..... g,h, SEM images of the D-nanotriskelions and the same sample stored

after 6 months. The nanotriskelions were stored in an environment of 0–5 °C. The morphology can be preserved well for at least 6 months.

We have added the description at the end of the paragraph right before Fig. 2.

“..... a few milliliters because of the involved complex chiral ligands or additional external stimuli^{4,5}. The Au nanotriskelions also show good stability. The morphology of the nanotriskelions can be preserved well for at least 6 months when they are stored in an environment of 0–5 °C (Supplementary Fig. 18g, h).”

Question 3: *It was very interesting that the 432 helicoid III and Au nanotriskelion had opposite chiroptical responses. However, in the paper, it was inferred only by the different structural characteristics of the two, which is too general an answer. It would be better if additional experiments were supported to reveal the mechanism.*

Response: We have carried out the CDS measurements on the 432 helicoid III NPs and Au nanotriskelions to study the CP-dependent scattering properties of the chiral NPs, as shown in the figure below. The scattering response of an individual NP can be correlated with its morphology. Although the 432 helicoid III NPs (grown from Au octahedrons) and the Au nanotriskelions (grown from Au nanodisks) are prepared in the presence of GSH with the same handedness, the nanotriskelions viewed from the $\langle 111 \rangle$ direction and the cubic helicoid NPs viewed from the $\langle 100 \rangle$ direction exhibit opposite geometrical chirality. As a result, they exhibit opposite chiroptical response in the wavelength range of 500–700 nm. For example, the L-432 helicoid III NPs exhibit an extinction g -factor of -0.088 at 590 nm and the average scattering g -factor is -0.45 at 640 nm. On the other hand, the L-nanotriskelions show an extinction g -factor of 0.05 at 595 nm and the average scattering g -factor is 0.57 at 650 nm. We believe that the combination of CD and CDS measurements is useful to comprehensively reveal the chiroptical behaviors of chiral plasmonic NPs.

Caption | Chiroptical response of the 432 helicoid III NPs and Au nanotriskelions. a, Constructed models and SEM images of the 432 helicoid III NPs. **b,** Extinction dissymmetry factor spectra measured from CD spectrometry for the 432 helicoid III NPs in solution. **c,** Average scattering *g*-factor spectra obtained from the CDS measurements on the 432 helicoid III NPs. **d,** Constructed models and SEM images of the Au nanotriskelions. **e,** Extinction dissymmetry factor spectra measured from CD spectrometry for the Au nanotriskelions in solution. **f,** Average scattering *g*-factor spectra obtained from the CDS measurements on the Au nanotriskelions.

The experimental results have been added as Supplementary Fig. 14, which is given in the response to Question 1 above. We have also added the relevant discussion in the second paragraph after Fig. 1.

“A combination of CD and CDS measurements is helpful in comprehensively revealing the relation between the helicoidal structure and the chiroptical response of chiral plasmonic nanoparticles. We next studied the differences in structural and chiroptical responses between the 432 helicoid III and Au nanotriskelions (Supplementary Figs. 13, 14). The 432 helicoid III nanocrystals with strong chiroptical properties were synthesized from Au octahedrons and are of a cubic geometry^{3,25}. Their six faces exhibit pinwheel-like patterns that consist of four highly curved arms of increasing widths. The strong chiroptical properties of the 432 helicoid III and Au nanotriskelions stem from their chiral geometries, which dominantly exhibit fourfold rotational (FFR) symmetry along the $\langle 100 \rangle$ directions and TFR symmetry along the $\langle 111 \rangle$ directions, respectively. Although the 432 helicoid III nanocrystals (grown from octahedrons) and the Au nanotriskelions (grown from nanodisks) are prepared in the presence of GSH with the same handedness, the nanotriskelions viewed from the $\langle 111 \rangle$ directions and the cubic helicoid nanocrystals viewed from the $\langle 100 \rangle$ directions exhibit opposite geometrical chirality (Supplementary Fig. 13). As a result, they exhibit opposite chiroptical responses in the wavelength range of 500–700 nm. For example, Supplementary Fig. 14 shows that the L-432 helicoid III exhibits an extinction *g*-factor of -0.088 at 590 nm and the average scattering *g*-factor is -0.45 at 640 nm. On the other hand, the L-nanotriskelions (Fig. 1c, f) show an extinction *g*-factor of 0.05 at 595 nm and the average scattering *g*-factor is 0.57 at 650 nm.”

REVIEWERS' COMMENTS

Reviewer #1 (Remarks to the Author):

The authors have adequately addressed my concerns, and the revised manuscript is suitable for publication in Nature Communications.

Reviewer #2 (Remarks to the Author):

Now it could be accepted.

Reviewer #3 (Remarks to the Author):

The authors carried out additional experiments to address the concerns and questions raised by the reviewers. The manuscript has been thoroughly revised and polished in a constructive way. Therefore, I recommend the publication of the current manuscript as it is. I appreciate the authors' effort and time for this work.

Reviewer #4 (Remarks to the Author):

The authors addressed my comments and the manuscript can now be accepted.

Reviewer #5 (Remarks to the Author):

I found additional experiments were well performed during the revision. But the study would be more insightful with the addition of the following data.

- 1) Although it has been claimed that stability was preserved for six months, quantitative information on the chirality of nanoparticles preserved for six months is also necessary.
- 2) Additional experiments have been requested by Au nanotriskelion and 432 Helicoid III to elucidate the mechanism of the opposing chiroptical response, but the response appears insufficient. The CDS and CD measurement findings were once more displayed in the case of the present response. It is necessary to further clarify the variables that oppose the two nanoparticles' chiropractic responses.

REVIEWERS' COMMENTS

Response to Reviewer #1

Comments: The authors have adequately addressed my concerns, and the revised manuscript is suitable for publication in Nature Communications.

Response: We thank this reviewer for the effort on evaluating our work, the highly positive comment, and the great help in the improvement of the manuscript.

Response to Reviewer #2

Comments: Now it could be accepted.

Response: We thank this reviewer for the effort on evaluating our work, the highly positive comment, and the great help in the improvement of the manuscript.

Response to Reviewer #3

Comments: The authors carried out additional experiments to address the concerns and questions raised by the reviewers. The manuscript has been thoroughly revised and polished in a constructive way. Therefore, I recommend the publication of the current manuscript as it is. I appreciate the authors' effort and time for this work.

Response: We thank this reviewer for the effort on evaluating our work, the highly positive comment, and the great help in the improvement of the manuscript.

Response to Reviewer #4

Comments: The authors addressed my comments and the manuscript can now be accepted.

Response: We thank this reviewer for the effort on evaluating our work, the highly positive comment, and the great help in the improvement of the manuscript.

Response to Reviewer #5

Comments: I found additional experiments were well performed during the revision. But the study would be more insightful with the addition of the following data.

Response: We thank this reviewer for the effort on evaluating our work, the highly positive comment, and the insightful questions and suggestions.

Question 1: Although it has been claimed that stability was preserved for six months, quantitative information on the chirality of nanoparticles preserved for six months is also necessary.

Response: We thank this reviewer for the valuable suggestion. We have provided the extinction g-factor spectra of the samples after synthesis and after storage for 6 months as Supplementary Fig. 17i.

Supplementary Fig. 17 | Scale-up synthesis and structural stability of the Au nanotriskelions. a–c, SEM images and extinction g-factor spectra of the L-nanotriskelions.....g–i, SEM images and extinction g-factor spectra of the D-nanotriskelions and the same sample stored after 6 months. The nanotriskelions were stored in an environment of 0–5 °C. The morphology (g,h) and chiroptical response (i) can be preserved well for at least 6 months.

Question 2: Additional experiments have been requested by Au nanotriskelion and 432 Helicoid III to elucidate the mechanism of the opposing chiroptical response, but the response appears insufficient. The CDS and CD measurement findings were once more displayed in the case of the present response. It is necessary to further clarify the variables that oppose the two nanoparticles' chiropractic responses.

Response: We thank this reviewer for the valuable suggestion. We are currently limited by our experimental conditions. We can only provide the experimental CDS and CD measurements to demonstrate the opposite chiroptical response of the Au nanotriskelions and 432 Helicoid III. We hope to be able to perform more experiments to understand the mechanism of the opposite chiroptical responses in the future. We are now trying to perform FDTD simulation and present the preliminary calculation results in the figure below. Figure a and b show the SEM images and measured extinction g-factor spectra of the Au nanotriskelions and 432 Helicoid III synthesized by L-GSH. On the L-type 432 helicoid III nanocrystals, the fourfold edge rotating clockwise around the center is observed along the <100> directions, while the threefold edge rotating counterclockwise is observed along the <111> directions on the L-nanotriskelions. The 432 helicoid III nanocrystals and nanotriskelions prepared in the presence of the same L-GSH display the opposite chiroptical properties. We use the constructed models with similar sizes to calculate the extinction spectra under the excitation of left-handed circularly polarized (LCP) and right-handed circularly polarized (RCP) light. The calculated extinction cross-sections under different excitation wavelengths are used to calculate the extinction g-factor, as described by

$$g_{\text{sim}} = 2 \times \frac{\sigma_{\text{LCP}} - \sigma_{\text{RCP}}}{\sigma_{\text{LCP}} + \sigma_{\text{RCP}}}$$

where σ_{LCP} and σ_{RCP} are the cross-sections for LCP and RCP incident light, respectively. The

simulated g -factor spectra show similar chiroptical properties to the experimental spectra. We hope that these simulation results are helpful for concluding that the opposite chiroptical properties of the Au nanotriskelions and 432 Helicoid III come from their structural difference. We are still working on the simulation to acquire more accurate results. The preliminary simulation results are therefore not presented in the revised manuscript.

Caption | Opposite chiroptical responses of the Au nanotriskelions and 432 Helicoid III.

a,b, SEM images and g -factor spectra of the Au nanotriskelions and 432 Helicoid III synthesized in the presence of L-GSH. **c**, Constructed models of the Au nanotriskelion and 432 Helicoid III. **d**, Simulated g -factor spectra of the constructed models.